# Controls on the formation and size of potential landslide dams and dammed lakes in the Austrian Alps

Anne-Laure Argentin[1], Jörg Robl[1], Günther Prasicek[1,2], Stefan Hergarten[3], Daniel Hölbling[4], Lorena Abad[4], and Zahra Dabiri[4]

[1]Department of Geography and Geology, University of Salzburg, 5020 Salzburg, Austria
[2]Interdisciplinary Center for Mountain Research, University of Lausanne, 1967 Bramois, Switzerland
[3]Institute of Earth and Environmental Sciences, University of Freiburg, 79104 Freiburg, Germany
[4]Department of Geoinformatics - Z_GIS, University of Salzburg, 5020 Salzburg, Austria

**Correspondence:** Anne-Laure Argentin (anne-laure.argentin@sbg.ac.at)

**Abstract.** Controls on landsliding have long been studied, but the potential for landslide-induced dam and lake formation has received less attention. Here, we model possible landslides and the formation of landslide dams and lakes in the Austrian Alps. We combine a slope criterion with a probabilistic approach to determine landslide release areas and volumes. We then simulate the progression and deposition of the landslides with a fluid dynamic model. We characterize the resulting landslide deposits with commonly used metrics, investigate their relation to glacial land-forming and tectonic units, and discuss the roles of the drainage system and valley shape. We discover that modeled landslide dams and lakes cover a wide volume range. In line with real-world inventories, we further found that lake volume increases linearly with landslide volume in case of efficient damming - when an exceptionally large lake is dammed by a relatively small landslide deposit. The distribution and size of potential landslide dams and lakes depends strongly on local topographic relief. For a given landslide volume, lake size depends on drainage area and valley geometry. Largest lakes form in glacial troughs, while most efficient damming occurs where landslides block a gorge downstream of a wide valley, a situation preferentially encountered at the transition between two different tectonic units. Our results also contain inefficient damming events, a damming type that exhibits different scaling of landslide and lake metrics than efficient damming, and is hardly reported in inventories. We assume that such events also occur in the real world and emphasize that their documentation is needed to better understand the effects of landsliding on the drainage system.

## 1 Introduction

Landslides are a major threat to human lives and infrastructure in mountain ranges worldwide. Beyond the direct hazard due to the moving mass, landslides can initiate natural hazard cascades by damming rivers and initiating catastrophic flash floods and debris flows (e.g. Costa, 1985; Costa and Schuster, 1988; Cui et al., 2009). Through such long-range effects, even unwitnessed landslides occurring in remote areas matter. Many landslide dams tend to fail shortly after their formation (Tacconi Stefanelli et al., 2015), while resistant dams get filled by sediments, complicating their documentation and the assessment of their impoundment potential. Thus, most landslide dam and lake inventories only contain relatively large dams. Several

geomorphometric indices have been developed to quantify the probability of landslides obstructing the valley and the stability of the resulting dams (Swanson et al., 1986; Canuti et al., 1998; Ermini and Casagli, 2002; Korup, 2004; Tacconi Stefanelli et al., 2016). However, studies on the formation of landslide dams and lakes, and on its dependence on factors that influence topography, such as contributing drainage area of rivers at their damming location, geologic preconditioning and long term climatic forcing are scarce.

Contributing drainage area at the damming position has been considered an important variable in computing obstruction and stability indices (e.g. Ermini and Casagli, 2002; Korup, 2004; Tacconi Stefanelli et al., 2016; Swanson et al., 1986). This attention to drainage area is due to the long term evolution of mountain landscapes: drainage area, as a proxy for discharge, is related to river flow length (Hack, 1957), channel slope (Flint, 1974) and river width (Finnegan et al., 2005; May et al., 2013). In particular, the latter two properties may exert a strong control on river damming by landslide deposits and on the volume of the thereby created lakes.

Mountain topography is conditioned by surface processes and the resistance of rocks against erosion. Both variables influence landslide occurrence (Hermanns and Strecker, 1999; Korup, 2008; Peruccacci et al., 2012), and likely exert control on dam and lake formation. Fluvial and glacial processes shape valleys and their flanks in typical ways. While fluvial valleys typically have a V-shaped cross-section with a narrow floor and straight flanks, glaciers scour U-shaped valleys with wide and flat valley floors and flanks steepening uphill (e.g. Davis, 1906; Harbor and Wheeler, 1992; Prasicek et al., 2015). Sediment filling, however, may cause widening of both glacial and fluvial valley floors (Schrott et al., 2003), and hanging sections of glacial valleys may exhibit inner gorges — very narrow fluvially incised canyons (Montgomery and Korup, 2011).

Rock strength constrains the steepness of hillslopes (Selby, 1982; Montgomery, 2001). Thus, lithology has an impact on the valley's morphology, influencing both the valley floor and the valley flanks (Robl et al., 2015; Goudie, 2016; Baumann et al., 2018). Landslides can effectively dam rivers in narrow valleys, since landslide volumes required to impound the river flow are small. However, only small lakes can form in narrow and steep valleys. Further, the steepness and relief of the valley flanks control the spreading of the landslide mass as well as its runout. Thus, both surface processes and lithology may influence the formation of landslide dams and lakes.

From these considerations, the question arises how potential landslide-dammed lakes are distributed across a mountain range, and how dam and lake characteristics are related and vary regionally as a function of drainage area, topography and rock type. While landslides and their occurrence have been extensively studied, supported by monitoring techniques ranging from remote sensing to geophysics (e.g. Nichol and Wong, 2005; Hölbling et al., 2012; Stähli et al., 2015), modeling of landslide distribution (Hergarten, 2012) and susceptibility (Reichenbach et al., 2018), potential damming of rivers by landslides and resulting lakes have received less attention (Korup, 2005).

In this study, we combine numerical methods from the field of natural hazards with concepts of long term landscape evolution. Therefore, we employ a modeling approach to investigate the influence of mountain topography, which differs in terms of predominant lithology and prevailing erosive surface processes (i.e. glacial versus fluvial conditions), on the potential occurrence of landslide dams and landslide-dammed lakes and on landslide and lake characteristics. We further calculate common

landslide dam obstruction and stability indices, develop a simple approach to estimate the volume of potential landslide-dammed lakes and compare our results to real-world inventories.

The Austrian Alps are a perfect natural laboratory to investigate the impact of differing landscape geometries on properties of potential landslide-dammed lakes. Beside the availability of a high resolution DEM (Open Data Österreich, starting 2015), a detailed geological map (Bousquet et al., 2012; Schmid et al., 2004) and an extensive landslide inventory (Kuhn, visited 2020.07.27), the study area features various topographic patterns related to contrasting lithological units and different climatic forcing (e.g. Robl et al., 2015). The topographic evolution of the Eastern Alps, of which the Austrian Alps are an essential part, started with the Late Oligocene - Early Miocene indentation of the Adriatic microplate into Europe (e.g. Handy et al., 2015). While timing and rates of topography formation of various parts of the Eastern Alps are still debated (see Bartosch et al., 2017, and references therein), north-south shortening and crustal thickening in concert with fluvial dissection by major alpine drainage systems (e.g. Inn, Salzach, Enns, Mur, Drau drainage systems) caused the formation of mountainous topography, with deeply incised valleys separated by interfluves with mountain peaks rising above 3 km. Located at mid-latitudes, the Austrian Alps were partly glaciated during the Pleistocene and still feature glaciers at the summit domains. While the topography in the western half of the study area was intensely reshaped by repeated glaciations, the eastern half shows a purely fluvial landscape (Fig. 3; Robl et al., 2008, 2015). Since major tectonic units with their characteristic lithological inventory strike west-east (Fig. 4; Bousquet et al., 2012; Schmid et al., 2004), we can directly compare the impact of glacial and fluvial dominated landscapes on occurrence and size of landslide dammed lakes within individual tectonic units. This allows a distinction between lithological and climatic control.

## 2 Materials and Methods

We use a novel combination of different numerical algorithms to model the formation of landslide dams and lakes. Our modeling workflow consists of three main steps: determination of landslide release areas and volumes, simulation of landslides, computation of geomorphometric parameters of landslide dams. Finally, we use the retrieved information to characterize and discuss dam and lake formation (Fig. 1).

### 2.1 Topographical, glacial and geological datasets

To model landslides we use a freely available LiDAR-based digital elevation model (DEM) of the Austrian Alps (Open Data Österreich, starting 2015) with a spatial resolution of $10\,\mathrm{m}$. The geophysical relief is based on the 1 arc second ASTER GDEM V3 (NASA/METI/AIST/Japan Spacesystems, and U.S./Japan ASTER Science Team, 2019). We use an Austrian landslide inventory containing the location of 194 events (Kuhn, visited 2020.07.27). We consider the glacially overprinted terrains to be found within the mapped extent of the last glacial maximum (LGM) originating from Ehlers and Gibbard (2004). We display the mapped tectonic units of the Alps (Fig. 4; Bousquet et al., 2012; Schmid et al., 2004) over the study area. However, the tectonic units are not homogeneous and comprise a high lithological and structural variability. Since lithology and discontinuities are a big control for erosion resistance, we do not venture to classify the tectonic units according to resistance to erosion.

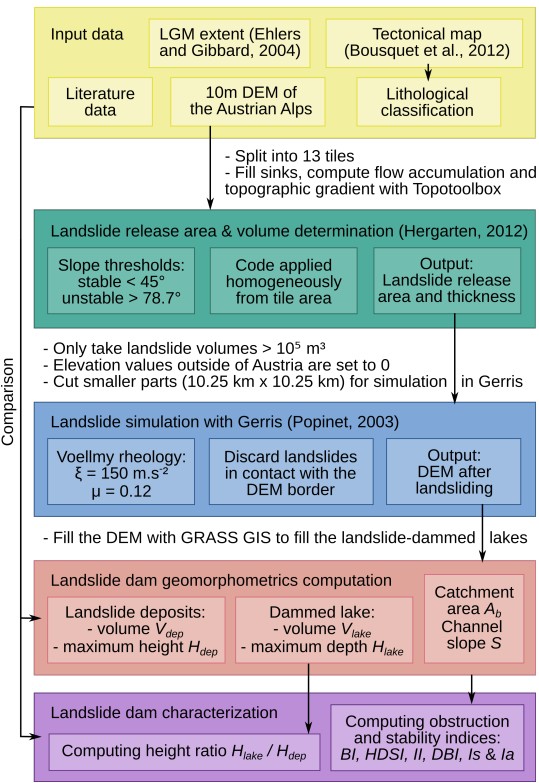

**Figure 1.** Workflow of modeled landslide dam creation across the Austrian Alps, and their geomorphometric analysis.

## 2.2 Geophysical relief

We computed the geophysical relief of the study region with a circular moving window of 2.5 km radius. The topographic envelope is obtained by taking the maximum elevation within the moving window. A Gaussian filter is applied to smooth the resulting dataset. Geophysical relief is then computed by subtracting the actual topography from the topographic envelope.

## 2.3 Determination of landslide release areas and volumes

Determining locations prone to landsliding and the respective potential volumes is challenging, in particular for landslides in
solid rock. The approach proposed by Hergarten (2012) still seems to be the only model which is able to predict the observed power-law distribution of rockfall and rockslide volumes in a simple and computationally efficient manner. The model is a combination of a geomorphometric analysis and a probabilistic approach. First, the algorithm stochastically chooses a seed pixel (i.e. a randomly picked pixel), then classifies the pixel slope to determine the stability of the local rock mass. Slope classification is based on lower and upper slope thresholds defining absolutely stable and absolutely unstable conditions,
respectively. A linear increase in the probability of failure is assumed between these two limits. In case of failure, material is removed from the destabilized pixel until its slope reaches the minimum slope threshold. This local change of topography

affects the slope of the adjacent pixels which are subsequently evaluated. In this way, the landslide area spreads until stable slope conditions at the seed pixel and its neighborhood are achieved. So the initiation of landslides depends on the local slope, while the final landslide size also depends on the size of sufficiently steep contiguous areas, which is related to the local relief.

For each seed pixel, the code finally outputs the area of the contiguous unstable pixels and the thickness of the substrate layer needed to be removed from each pixel to stabilize the area. In the next step, this data is used as release area and volume to model the landslides.

Hergarten (2012) found that the exponent of the landslide size distribution shows only a weak dependence on the threshold slopes $s_{\min}$ and $s_{\max}$, while the total number of events triggered and the maximum event size are strongly affected by these

parameters. It can be expected that $s_{\min}$ and $s_{\max}$ depend on lithology. However, the dependency has not been investigated systematically so far. Hence, we use the same uniform slope threshold values, $s_{\min} = 1$ (45°) and $s_{\max} = 5$ (79°), applied by Hergarten (2012) to reproduce the distribution of landslide volumes in the Alps. Implications on landslide metrics and their spatial distribution are explained in detail in the Discussion section.

To avoid memory issues in the simulations, we split the DEM into 14 smaller tiles for computational reasons and introduce

buffer frames to account for the run-out of the landslides. We fill the sinks of the DEM and compute the flow accumulation and topographic gradient using Topotoolbox (Schwanghart and Kuhn, 2010; Schwanghart and Scherler, 2014).

## 2.4    Landslide simulation

Once the landslide release volumes have been determined, we simulate the runout of the landslides. As the model for the volume involves no time scale, it is assumed that the entire volume is released instantaneously.

We use a depth-averaged granular flow similar to shallow-water equations as introduced by Savage and Hutter (1989) in combination with the Voellmy rheology. In comparison with frictional and Bingham rheologies, the Voellmy rheology most accurately reproduces the debris deposition when simulating landslides with depth-averaged flow solvers (Hungr and Evans, 1996). This rheological model makes use of two parameters (Voellmy, 1955): a velocity squared drag coefficient $\xi$ (consisting of density and drag coefficient) and a dry friction coefficient $\mu$ (the ratio between the needed sliding force and the force

perpendicular to the rupture surface). Drag increases with velocity. Hungr and Evans (1996) found values of $\xi$ ranging from 100 to 1000 $\mathrm{ms}^{-2}$, and values of $\mu$ from 0.03 to 0.24 by back-analyzing 23 rock-avalanches. An analysis using Gerris with the Voellmy rheology on the 1987 Val Pola rock avalanche in Italy found that $\xi = 150\,\mathrm{ms}^{-2}$ and $\mu = 0.12$ are the most appropriate coefficients (Sanne, 2015).

Testing the influence of the two parameters, we found that they show no consistent influence on the modeled lake volume

results (Supplementary Fig. A1). While the velocity squared drag coefficient $\xi$ has only a slight negative impact on landslide deposit height, an increase in dry friction $\mu$ results - as expected - in notably higher values (Supplementary Fig. A1b). However, neither $\xi$ nor $\mu$ does systematically change lake depths and volumes (Supplementary Fig. A1a). This shows that, while maximum deposit heights increase, depths and volumes of dammed lakes and hence average geometries of landslides damming valleys are not consistently affected. Thus, we chose to keep the Voellmy coefficients determined by Sanne (2015). We do not

take into account the entrainment of sediments and the loosening of bedrocks, that could increase the volume of the detached mass.

Several methods and various software tools are currently available to implement depth-averaged flows and model flow slides, debris flows and avalanches and reconstruct landslide dams (Hussin et al., 2012; Schraml et al., 2015; Delaney and Evans, 2015; Lin and Lin, 2015). We use Gerris because of its computational performance, flexibility, widespread use in fluid-flow mechanics, and its open-source policy (Popinet, 2003). Gerris can be employed to simulate avalanches and debris flows even in steep terrain due to a series of correction terms, which allow to bypass the almost-horizontal fluid table requirement by solving the shallow water equations in Cartesian coordinates (Hergarten and Robl, 2015). Correction terms for the acceleration of the fluid layer and the applied flow resistance law (Voellmy rheology) were tested and validated against Rapid Mass Movement Simulation (RAMMS), the leading software and industry standard for rapid mass movement simulation (e.g. Christen et al., 2010).

To reduce computation time, we discard landslides with volumes $<10^5$ m$^3$. We assume sea level altitude (i.e. 0 m elevation) outside of Austria. This affects the flow simulation and we thus discard manually the 77 landslides and lakes in contact with the DEM border. As such, there is an underestimated landslide dam density within 8 km of the DEM border. We model each landslide for a run-out time of six minutes. Due to high flow velocities, this time span is sufficiently long for the rock mass to deposit (i.e. for the landslide momentum to decrease to a small fraction of its maximum values).

After completing the simulation, the landslide mass is added to the DEM. The DEM is then filled using GRASS GIS and the maximum landslide-dammed lake volume is computed by subtracting the original DEM from the filled DEM including the landslide mass.

## 2.5 Geomorphometric parameters, damming percentage and indices of landslide dams

We compare the geomorphometric parameters (Table 1) of our modeled landslide dams to those of landslide dams from existing inventories (Table 2). Except for Fan et al. (2012) and Tacconi Stefanelli et al. (2015), these studies focus on river-damming landslides only. Various indices have been developed to predict the ability of a landslide to dam a valley and the longevity of the dam. Those indices rely on simple parameters of the landslide, dam, lake and valley: the landslide dam volume $V_{dam}$ (m$^3$) and height $H_{dam}$ (m), the landslide volume $V_{landslide}$ (m$^3$), the lake volume $V_{lake}$ (m$^3$), the upstream catchment area $A_b$ (km$^2$) and the local slope of the fluvial channel at the point of damming $S$ (m/m). They allow to estimate the potential landslide damming risk.

To characterize our modeled dams, we use the landslide deposit volume $V_{dep}$ and the upstream catchment area of the dam-covered pixel with the highest flow accumulation ($A_b$). The slope $S$ is taken as the D8 slope (steepest outwards slope for a grid cell to one of its eight neighbors) at the same pixel location. Two metrics can be considered as proxies for $H_{dam}$: the maximum height of the landslide deposit $H_{dep}$ (m) and the maximum depth of the dammed lake $H_{lake}$ (m) (Fig. 2). Taking $H_{lake}$ as proxy for $H_{dam}$ is possible because we use a filled, and hence depression-free DEM, as a basis for landslide modeling. The maximum depth of the lake must thus be located close to the dam and represents the vertical distance from the lowest point in the dam cross-section (Fig. 2b) to the lowest point in the valley longitudinal view (Fig. 2c). In contrast, $H_{dep}$ is located in the

**Table 1.** Geomorphometric parameters mentioned in the article and their notation.

*° The extent of the sediments involved in the dam is hardly definable, thus the dam volume is not computed.*

| | |
|---|---|
| $V_{landslide}$ | Landslide volume |
| $V_{dam}$° | Dam volume |
| $V_{dep}$ | Volume of landslide deposit |
| $V_{lake}$ | Volume of landslide-dammed lake |
| $H_{dam}$ | Dam height (cf. Fig. 2) |
| $H_{dep}$ | Maximum landslide deposit height, "dam height proxy" (cf. Fig. 2) |
| $H_{lake}$ | Maximum dammed lake depth, "dam height proxy" (cf. Fig. 2) |
| $A_b$ | Catchment area upstream of dam blockage |
| $S$ | Channel slope at the dam pixel of highest flow accumulation |
| $L_{lake}$ | Lake length (along the river) |
| $W_{lake}$ | Lake width (cross-sectional) |
| $V_{p\ lake}$ | Predicted volume of landslide-dammed lake using easily calculable geomorphic parameters. |

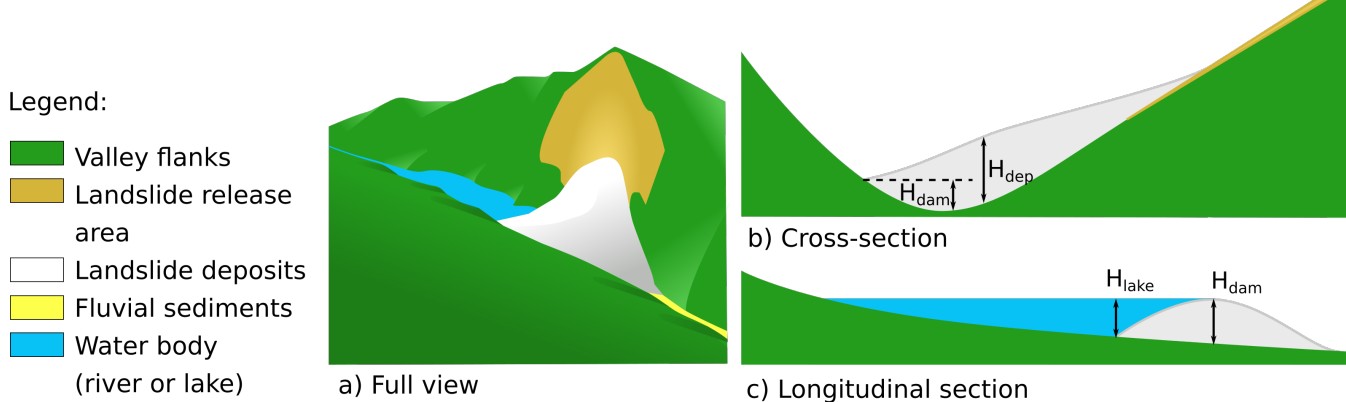

**Figure 2.** Definition of the heights $H_{lake}$, $H_{dam}$ and $H_{dep}$ in cross and longitudinal sections of a landslide dam. $H_{lake}$ and $H_{dep}$ can be easily computed while $H_{dam}$ cannot. $H_{lake}$: maximum lake depth, $H_{dam}$: landslide dam height, $H_{dep}$: maximum landslide deposit height.

deposit but not necessarily close to the dam (Fig. 2b). We assume the height metrics to follow the relation:

170 $$H_{lake} \leq H_{dam} \leq H_{dep} \tag{1}$$

Landslide dams are commonly classified in a binary and simple fashion between complete and partial blockages based on their planform geometry (Hermanns, 2013). Complete dam blockages are landslide deposits that fully obstructed the river flow and formed a lake. Partial dam blockages are landslide deposits that encountered the river bed and may have triggered an avulsion, but did not completely impound the river. Complete blockages are much more dangerous than partial blockages

**Table 2.** Landslide dam and lake volume ranges from around the world compared to our generated landslide-dammed lakes. The Wenchuan landslide dams all originate from the 2008 Wenchuan earthquake. Numbers are approximates.

[a] *Modeled landslide dams and lakes.* [b] *The modeled landslides with volume below $10^5$ m³ were not computed.* [c] *The $H_{dam}$ proxies are written $H_{lake}$ | $H_{dep}$.* [d] *Except for the Tangjiashan landslide dam outlier which impounded $3 \times 10^8$ m³ of water.* [e] *Usoi dam, lake Sarez, Tajikistan.* [f] *Number of database events with provided lake volume.*

| Area & Reference | Min $V_{landslide}$ or $V_{dam}$ (m³) | Max $V_{landslide}$ or $V_{dam}$ (m³) | Min $V_{lake}$ (m³) | Max $V_{lake}$ (m³) | Min $H_{dam}$ (m) | Max $H_{dam}$ (m) | Damming landslide number |
|---|---|---|---|---|---|---|---|
| **Alps, Austria [a] (This paper)** | **$7.7 \times 10^4$** [b] | **$9.9 \times 10^7$** | **0.0** | **$7.9 \times 10^7$** | **0 | 3** [c] | **75 | 155** [c] | 1057 |
| Alps, Austria (Dufresne et al., 2018) | $1.5 \times 10^7$ | $2.1 \times 10^9$ | 0.0 | $1.1 \times 10^9$ | 40 | 450 | 5 |
| Apennines, Italy (Tacconi Stefanelli et al., 2016) | $3.0 \times 10^4$ | $1.1 \times 10^8$ | - | - | - | > 100 | 300 |
| Taiwan (Chen et al., 2014) | $6.0 \times 10^2$ | $5.0 \times 10^8$ | - | - | 3 | 300 | 64 |
| Wenchuan, China (Fan et al., 2012) | - | $7.5 \times 10^8$ | $4.2 \times 10^3$ | $2.1 \times 10^7$ [d] | 1 | 160 | 828 |
| New Zealand (Korup, 2004) | $4.0 \times 10^4$ | $2.7 \times 10^{10}$ | $1.0 \times 10^4$ | $5.0 \times 10^9$ | 5 | 800 | 232 |
| Japan (Korup, 2004) | $3.0 \times 10^3$ | $1.2 \times 10^9$ | $2.0 \times 10^3$ | $6.0 \times 10^8$ | - | - | |
| USA (Korup, 2004) | $1.9 \times 10^3$ | $1.5 \times 10^9$ | $1.0 \times 10^3$ | $5.5 \times 10^8$ | - | - | |
| World-wide (Korup, 2004) | $4.3 \times 10^3$ | $1.3 \times 10^9$ | $2.0 \times 10^3$ | $4.0 \times 10^9$ | - | - | 184 |
| World-wide (Costa and Schuster, 1988) | $7.0 \times 10^4$ | $2.8 \times 10^9$ | $1.1 \times 10^5$ | $6.8 \times 10^8$ | 3 | 550 | 225 |
| World-wide (Fan et al., 2020) | $1.2 \times 10^3$ | $5.0 \times 10^9$ | 0.0 | $1.6 \times 10^{10}$ [e] | 2 | 1000 | 443 [f] |

and tend to trap sediments while partial dams increase the river sediment load. Following Croissant et al. (2019), we assume that all of our modeled landslides, given their high volume, the initiating slope threshold and the self-similar structure of river networks, reach a river bed, and thus qualify as either complete or partial blockages (Lucas et al., 2014). However, to avoid differentiating binarily between complete and partial dams through a visual inspection of thousands of modeled landslide dams, we compare $H_{dep}$ to $H_{lake}$ by using the $\frac{H_{lake}}{H_{dep}}$ ratio to create a continuous damming scale. If $\frac{H_{lake}}{H_{dep}}$ is small, then $H_{dep} \gg H_{lake}$, the landslide likely did not fully obstruct the valley, while if $\frac{H_{lake}}{H_{dep}}$ is closer to 1, $H_{dep} \approx H_{lake}$, the landslide probably obstructed the valley.

In our study, we compare six obstruction and stability indices. Obstruction criteria have been developed to differentiate landslides leading to complete blockages from those leading to partial ones, while stability criteria aim to assess dam stability (e.g. the probability of the dam not failing) from simple geomorphometric parameters. Some indices can serve as both obstruction and stability criteria. The two indices that aim to classify the landslides according to their potential obstruction power and

stability are the Blockage Index $BI$ and the Hydromorphological Dam Stability Index $HDSI$. The $BI$

$$BI = \log\left(\frac{V_{dam}}{A_b}\right) \tag{2}$$

which divides landslide dam volume by the upstream catchment area, was developed by Swanson et al. (1986), then modified by Canuti et al. (1998), who replaced the landslide volume by landslide dam volume. Tacconi Stefanelli et al. (2016) introduced more recently the $HDSI$

$$HDSI = \log\left(\frac{V_{landslide}}{A_b S}\right) \tag{3}$$

which differs from the $BI$ by taking into account the channel slope. Both indices can be computed prior to landsliding (using the original version of the $BI$).

Conversely, all other four indices use geomorphometric parameters linked to the dam or/and the lake, and thus can only be used after landsliding to assert the dam stability. Casagli and Ermini (1999) proposed the Impoundment Index $II$

$$II = \log\left(\frac{V_{dam}}{V_{lake}}\right) \tag{4}$$

which accounts for lake volume when estimating the landslide dam stability. The Dimensionless Blockage Index $DBI$

$$DBI = \log\left(\frac{A_b \cdot H_{dam}}{V_{dam}}\right) \tag{5}$$

coined by Ermini and Casagli (2002), considers the dam height, allowing to indirectly take into account the steepness of the dam flanks. Korup (2004) introduced two new indices also based on landslide dam height, the Backstow Index $Is$ and the Basin Index $Ia$

$$Is = \log\left(\frac{H_{dam}^3}{V_{lake}}\right), \quad Ia = \log\left(\frac{H_{dam}^2}{A_b}\right) \tag{6}$$

In contrast to the $BI$ and $HDSI$, the stability indices ($II$, $DBI$, $Is$ and $Ia$) use a non-dimensional combination of properties (volume per volume, or area per area), which should give more consistent results across different scales.

While the indices $BI$, $II$, and $DBI$ use the volume of the dam instead of the total volume of the deposits, determining $V_{dam}$ automatically for large data sets is nontrivial. We therefore use $V_{dep}$ instead of $V_{dam}$ when computing the indices. This may lead to an overestimation of the volume if significant parts of the deposits do not reach the valley floor.

In turn, $V_{dep}$ is in general underestimated by our approach, mainly because the increase in volume by bulking via fragmentation and entrainment of further material is not taken into account. The Gerris solver even loses a small part of the volume at the tail of the landslide since layers below a given threshold thickness are disregarded. Thus, we have the following relationship: $V_{dam} \leq V_{dep} < V_{landslide}$. However, the underestimation of $V_{dep}$ is only relevant if we consider the landslide dam in relation to the detached volume which is not a subject of this study.

## 3 Results

We calculated landslide release areas with 100 landslide seeds per $\text{km}^2$ and obtained 1057 release volumes larger than $10^5 \ \text{m}^3$ in the Austrian Alps. We then used these release volumes and simulated the runout of landslides. We further investigated if

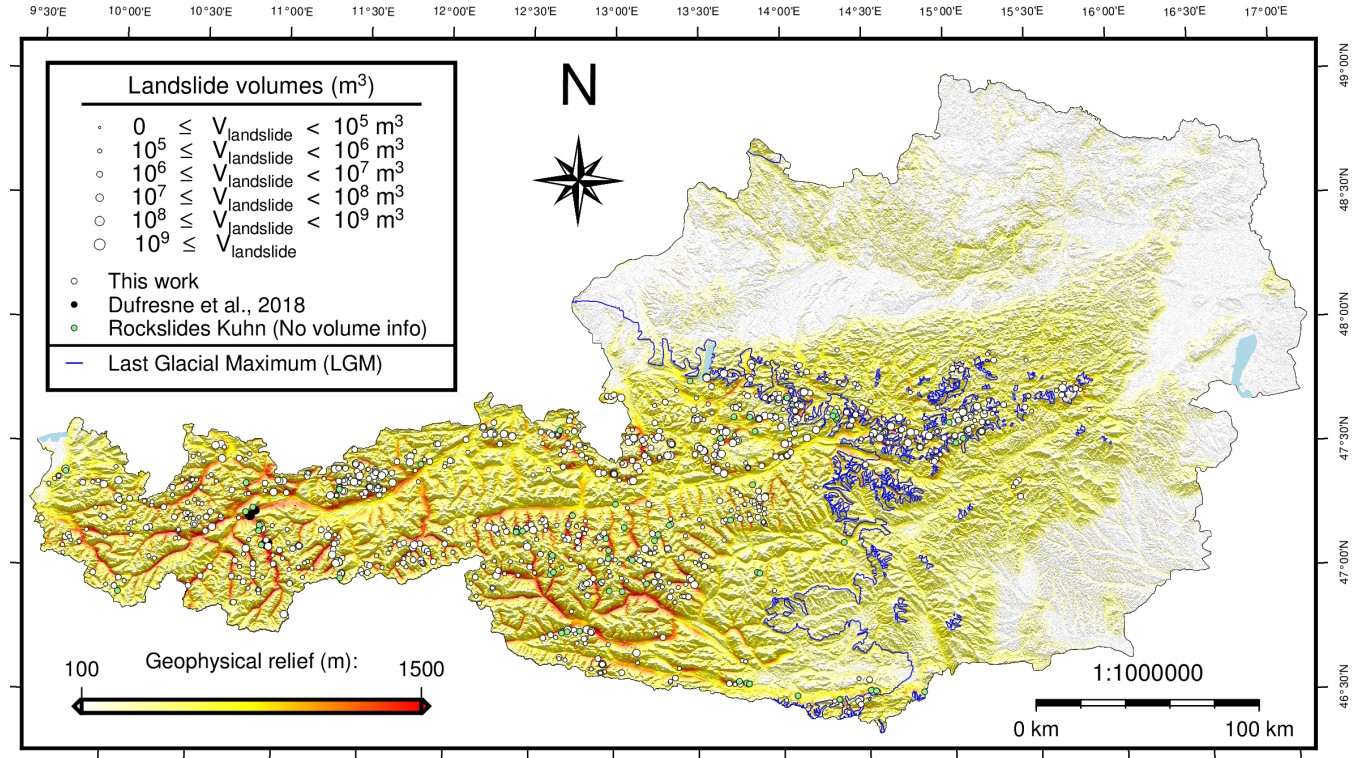

**Figure 3.** Spatial distribution of modeled and real-world landslides in the Austrian Alps plotted on geophysical relief. Landslide volume is reflected by the circle size. LGM extent is depicted by a blue line (Ehlers and Gibbard, 2004). The landslides marked by the green circles were compiled by Kuhn (visited 2020.07.27). Hillshades were computed from freely available LiDAR-based digital elevation model (DEM) of the Austrian Alps (Open Data Österreich, starting 2015).

landslide-dammed lakes are formed. In the following result sections, we describe the 1057 simulated landslides and landslide-dammed lakes: their spatial distribution, their geometric characteristics and their associated stability and obstruction indices.

### 3.1 Distribution of simulated landslides and landslide dams across the Austrian Alps

The distribution of reported landslides in the Austrian Alps (Kuhn, visited 2020.07.27; Dufresne et al., 2018) is linked to
220  topographic characteristics and geomorphological process domains (Fig. 3, green circles). Most of the landslides are located in the western part of the study region, within high topography with significant relief occupied by glaciers during the last glacial maximum (LGM). Modeled landslides (Fig. 3, white circles) and inventory landslides (Fig. 3, green circles) show similar spatial patterns, thus implying that the spatial heterogeneity in landslide occurrence arises from landscape characteristics. High local slope has a strong positive influence on simulated landslide density, while high landslide volume is rather driven by high
225  relief.

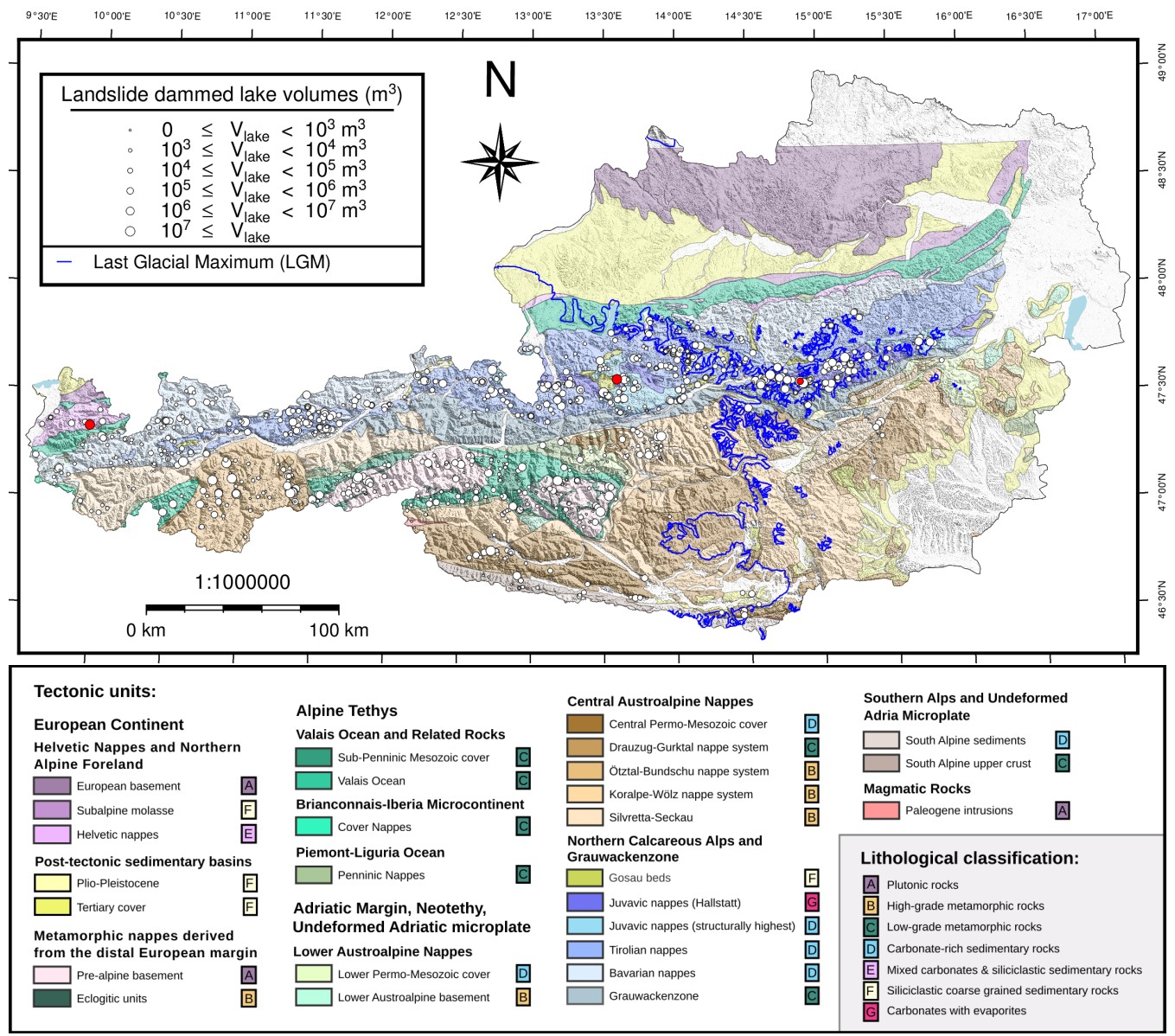

**Figure 4.** Spatial distribution of modeled landslide-dammed lakes in the Austrian Alps plotted on a map of tectonic units modified after Bousquet et al. (2012) (see also Schmid et al., 2004). The landslide-dammed lake volume is indicated by circle size. LGM extent is depicted by a blue line (Ehlers and Gibbard, 2004). Hillshades were computed from freely available LiDAR-based digital elevation model (DEM) of the Austrian Alps (Open Data Österreich, starting 2015). The three landslide-dammed lakes highlighted in red are mentioned in the text.

**Table 3.** Landslide dam statistics for glacial and fluvial terrain.

| Imprint | Glacial | Fluvial |
|---|---|---|
| Area (km$^2$) | 33751 | 45643 |
| Number of landslides | 999 | 58 |
| Landslide density (km$^{-2}$) | $3.0{\times}10^{-2}$ | $1.3{\times}10^{-3}$ |
| Mean deposit volume (m$^3$) | $8.6{\times}10^6$ | $3.1{\times}10^6$ |
| Mean lake volume (m$^3$) | $1.5{\times}10^6$ | $5.9{\times}10^5$ |
| Mean of the $H_{lake}$ / $H_{dep}$ | 0.26 | 0.39 |
| Mean of the $V_{lake}$ / $V_{dep}$ | 0.15 | 0.25 |

Areas with high and low geophysical relief values spatially coincide with contrasting tectonic units (compare Figs. 3 and 4). This suggests that lithology exerts an important control on geophysical relief and hence landslide occurrence in the study region (Fig. 4). For example, major historical landslides are reported for the Northern Calcareous Alps (NCA) but not for the adjacent Greywacke zone (the structural base of the NCA). This distribution is mimicked by our model due to the contrasting relief and slope characteristics of the two lithological units. Similarly, the prediction of many large landslides in the Ötztal-Bundshu nappe system and the Pre-alpine basement (gneisses of the Tauern Window) is consistent with landslide occurrence in the landslide inventory (Kuhn, visited 2020.07.27), while a significantly lower tendency to landsliding is both modeled and reported in nearby tectonic units (e.g. Silvretta-Seckau or Koralpe-Wölz nappe system).

Glacial erosion is known to increase valley relief and to steepen valley flanks (Shuster et al., 2005; Valla et al., 2011). To further investigate the role of glacial imprint in preconditioning the occurrence of modeled landslides, we computed landsliding densities and spatially distinguished $\frac{H_{lake}}{H_{dep}}$ ratios (Table 3). 94.5% of the predicted landslide release areas are situated in glacially overprinted terrain. The glacial and fluvial landslide densities are $3.0{\times}10^{-2}$ and $1.3{\times}10^{-3}$ landslides per km$^2$, respectively. As expected, the disparities in landslide occurrence in glacial and fluvial terrain are even stronger for very large landslides. This is reflected in the mean volume that is about 2.8 times higher in the glacially overprinted domain than in the fluvial area. The large landslide volumes also result in larger lake volumes. On average, these are about 2.5 times higher in the glacially overprinted areas. In relation to the deposit volume, the lake volume is, however, slightly smaller in the glacially overprinted areas, indicating that smaller lakes are dammed by a landslide deposit of a given volume. The same applies to lake depths and deposit depths. Both effects are probably a consequence of differences in glacial and fluvial valley geometry.

### 3.2 Comparison of geomorphometric parameters

We first compared deposit volumes $V_{dep}$, volumes of the dammed lakes $V_{lake}$ and dam heights $H_{lake}$ and $H_{dep}$ of our modeled landslide dams to landslide inventories (Table 2). The modeled deposit volumes $V_{dep}$ range from the defined minimum of $10^5$ m$^3$ to a maximum of almost $10^8$ m$^3$, while the lake volumes $V_{lake}$ range from 0 to $7.9 \times 10^7$ m$^3$. Both the $V_{dep}$ and the $V_{lake}$ maximums are 10 times smaller than the biggest dam and lake volume reported in Austria, and between 10 and 100 times lower than the largest volumes found in Japan, the USA and New Zealand. This finding is not particularly surprising as the

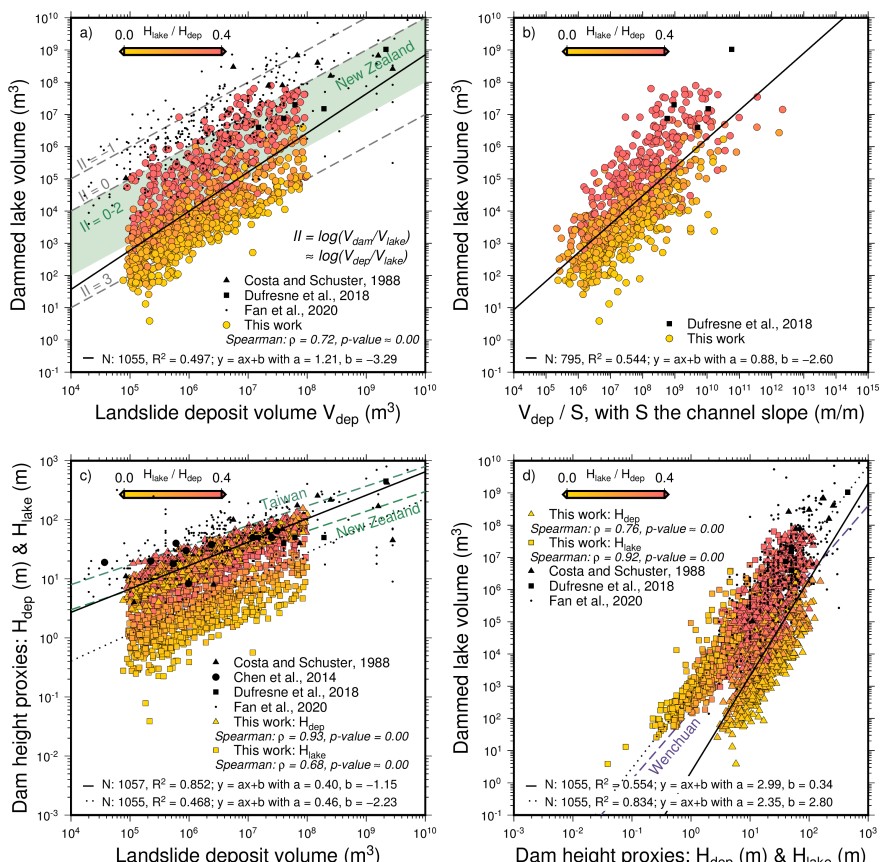

**Figure 5.** Bi-logarithmic diagrams of the landslide dam and lake metrics: (a) dammed lake volume in relation to landslide deposit volume (a.k.a. Impoundment Index) $II$, (b) dammed lake volume vs. channel slope, (c) landslide dam height proxies vs. landslide deposit volume, (d) landslide dam height proxies vs. dammed lake volume. $\frac{H_{lake}}{H_{dep}}$ is color-coded. $a$ and $b$ represent slope and intercept of the fitted power-laws, respectively. $N$ varies as 2 landslides did not dam a lake and channel slopes equal to zero where not considered. New Zealand data from Korup (2004), Taiwan data from Chen et al. (2014), Wenchuan data from Fan et al. (2012), and world-wide data from Fan et al. (2020).

250   potential for very large landslides decreases through time after deglaciation (Hergarten, 2012). The simulated volume ranges are further in accordance with landslide dam and lake volumes found in the Apennines by Tacconi Stefanelli et al. (2016). The maximum of our $H_{lake}$ proxy for landslide dam heights is 6 times lower than reported for Austria, 10 times lower than in New Zealand, and 2 times lower than those from Wenchuan and Italy. However, the maximum of our $H_{dep}$ proxy is similar to those from Wenchuan and Italy.

255   The introduced geomorphometric parameters show distinct relationships (Fig. 5), which have also been identified in inventories. We carried out Spearman correlations and fitted power-law relations between the considered properties. Although the modeled deposit and lake volumes are strongly correlated, with a Spearman-$\rho$ of 0.72 (Fig. 5a), the deposit volume can only explain a part of the variability in the lake volume dataset, with a coefficient of determination ($R^2$) of 0.497. The $II$,

the logarithm of $\frac{V_{dep}}{V_{lake}}$, of the modeled landslide dams stretches from 0 to 3, while values from literature are mostly found between 0 and 2 in Austria (Dufresne et al., 2018) and New-Zealand (Korup, 2004), and between -1 and 1 for largest dams world-wide (Costa and Schuster, 1988; Fan et al., 2020) (Fig. 5a). The height ratio $\frac{H_{lake}}{H_{dep}}$ of our modeled landslides is strongly correlated to the $II$ (color coding in Fig. 5), and field observations of landslide dams are found among the simulated results with high height ratios. In this way, $\frac{H_{lake}}{H_{dep}}$ is linked to $\frac{V_{dep}}{V_{lake}}$, and both ratios are indicators for efficient damming, i.e. relatively small landslides damming relatively large lakes. Power-law fitting shows that lake volume increases non-linearly with deposit volume for all events and that the mean $II$ decreases from 2.2 to 1.6 over the considered volume range. For damming events with highest lakes volumes relative to deposit volumes, i.e. efficient damming, however, lake volume increases linearly with deposit volume.

Lake volume exhibits an inverse relationship with channel slope. Combining the channel slope (Fig. 5b) with deposit volume explains more of the lake volume variability ($R^2 = 0.544 > R^2 = 0.497$).

The dam height proxies $H_{dep}$ and $H_{lake}$ scale non linearly with the deposit volume (Fig. 5c), reproducing reported relationships (Costa and Schuster, 1988; Chen et al., 2014; Dufresne et al., 2018). The deposit height correlates strongly ($\rho = 0.93$) and presents less dispersion than the lake depth ($\rho = 0.68$). Similar to the deposit to lake volume relation, the lake depth fits the literature data best for high $\frac{H_{lake}}{H_{dep}}$ ratios. The power law exponents ($\alpha = 0.40$, $\alpha = 0.46$) are close to each other. Landslides of volumes smaller than $10^6$ m$^3$ show a power-law of exponent $\alpha = 0.448$ when fitted separately, while landslides with volumes larger than $10^7$ m$^3$ give a power-law of exponent $\alpha = 0.325$.

The lake volume scales non-linearly with the dam height proxies $H_{dep}$ and $H_{lake}$ (Fig. 5d). The situation is reversed to Fig. 5c, such that the lake depth correlates strongly with the lake volume ($\rho = 0.92$), which conforms to the trends in inventories. The deposit height shows a weaker correlation with lake volume ($\rho = 0.76$). In both cases, dams and lakes with similar $H_{lake}$ and $H_{dep}$, thus high $\frac{H_{lake}}{H_{dep}}$ ratios, match the field observations better.

The lake depth scales non linearly with the deposit height (Supplementary Fig. B1), with similar coefficients and behavior than found with the lake and deposit volumes.

### 3.3 Obstruction and stability indices

We apply six obstruction and stability indices to our modeling results (Fig. 6). Korup (2004) and Tacconi Stefanelli et al. (2015) determined index thresholds, which separate their landslide dams into different obstruction and stability classes:

– No data: no partial or complete landslide dams were observed.

– Partial: the landslides obstructed only partly the river bed to form a partial dam.

– (Complete-) Unstable: the landslides obstructed fully the river bed, but the formed dams breached catastrophically.

– (Complete-) Stable: the landslides obstructed fully the river bed, and the formed dams did not experience any catastrophic failure. However, they may have disappeared by sediment infilling or gradual incision.

– Undefined: the landslide dams are either partial, (complete-)unstable or (complete-)stable.

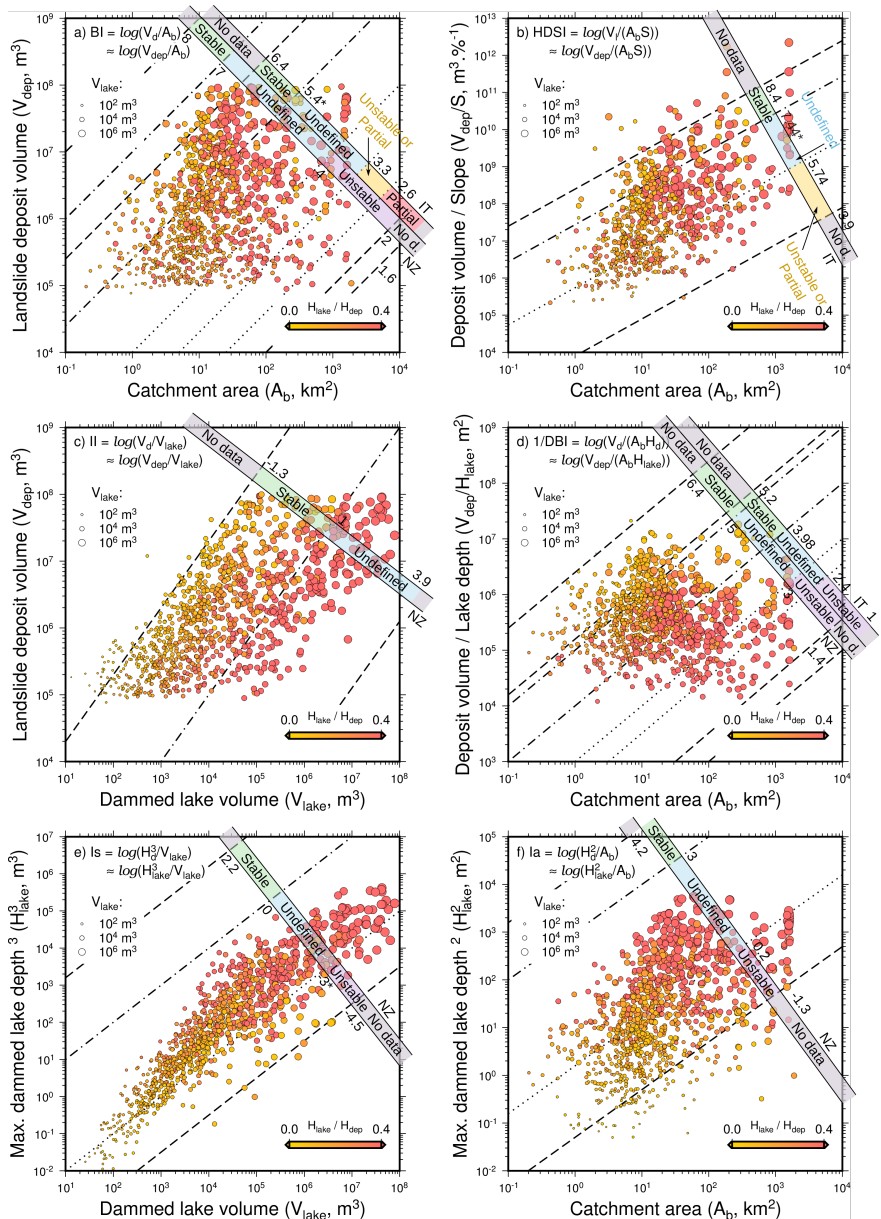

**Figure 6.** Bi-logarithmic diagrams of landslide dam classification according to two obstruction and stability indices, (a) the Blockage Index $BI$ and (b) the Hydromorphological Dam Stability Index $HDSI$, and four stability indices, (c) the Impoundment Index $II$, (d) the Dimensionless Blockage Index $DBI$, (e) the Backstow Index $Is$ and (f) the Basin Index $Ia$. Circle color represents $\frac{H_{lake}}{H_{dep}}$ and circle size depicts lake volume. The obstruction and stability ranges from literature are indicated by scales, with the threshold values annotated. Threshold lines are dashed for "No Data", dot-dashed for "Stable", dotted for "Unstable". New Zealand data (Korup, 2004) is indicated by NZ and Apennines data from Italy (Tacconi Stefanelli et al., 2016) by IT. The threshold values marked with an asterisk present a few outliers in the reported literature data. The cluster of values with a catchment area of $10^3$ km$^2$ are located in the same area in the Gesäuse mountain range, in the Enns catchment.

We compared our modeled dams and related lakes to their obstruction and stability classes (Fig. 6). Our dams fall into different fields, depending on the applied indices.

For the $BI$, Korup (2004) and Tacconi Stefanelli et al. (2015) studied the Southern Alps, New Zealand and Apennines, Italy, respectively, and found different limits for the stability classes. This affects the stability classification of our dams (Fig. 6a). Many modeled dams are considered stable in the Apennines classification scheme, while none are stable according to the New Zealand scheme. The relation between $BI$ and $\frac{H_{lake}}{H_{dep}}$ is ambiguous, but we observe that $\frac{H_{lake}}{H_{dep}}$ and $V_{lake}$ are positively correlated with catchment area $A_b$.

The $HDSI$, originally defined for the Apennines (Tacconi Stefanelli et al., 2015), presents no obvious relation to the $\frac{H_{lake}}{H_{dep}}$ ratio. Our data range is more extended than determined for the Apennines (Fig. 6b). Again, a minority of dams is considered stable in the $HDSI$, while the majority falls into the undefined class and a considerable fraction is classified unstable or partially stable.

For the $II$ (Fig. 6c), the majority of landslides, in particular those with small lake volumes, fall in the stable class as determined for the Southern Alps, with the tendency of stability to decrease with lake volume. Further, the $II$ displays a strong positive correlation with the $\frac{H_{lake}}{H_{dep}}$ ratio and lake volumes.

For the $DBI$, the situation is similar to the $BI$, with mountain range-dependent class definitions and no overlap between the stable classes (Fig. 6d). Accordingly, our modeled dams can either be classified stable or undefined or even undefined or unstable. The $DBI$ shows a strong positive correlation with the $\frac{H_{lake}}{H_{dep}}$ ratios. High lake volumes tend to gather around medium $DBI$ values.

According to the $Is$ classification from the Southern Alps, our modeled lakes are either classified undefined or unstable, with no lakes in the stable class. Further, The $Is$ presents no correlation with the $\frac{H_{lake}}{H_{dep}}$ ratio (Fig. 6e).

The $Ia$ classes determined in the Southern Alps (Fig. 6f) lead to our modeled lakes being classified either undefined or unstable and far from stable. The relations between $Ia$ and $\frac{H_{lake}}{H_{dep}}$ ratio and lake volumes are ambiguous.

Summing up, the predictions on the stability of our modeled landslide dams vary strongly depending on the indices and thresholds chosen (e.g. $II$, $Ia$). Further, the indices display changing correlations with the $\frac{H_{lake}}{H_{dep}}$ ratio, a proxy for efficient damming. While the $II$ and $DBI$ both link low $\frac{H_{lake}}{H_{dep}}$ ratios with high stability results, the other four indices show no obvious relationship. The $\frac{H_{lake}}{H_{dep}}$ ratio is correlated positively with the catchment area $A_b$, the lake volume $V_{lake}$ and height $H_{lake}$, with higher values for bigger catchments, but do not display any obvious correlation with the deposit volumes $V_{dep}$ and their slope $V_{dep}/S$.

There are no big trends linked to tectonic units in the indices plots (Supplementary Fig. C1). Tectonic units are homogeneously distributed in the $BI$ plot, except for the Juvavic nappes (Hallstatt), which present slightly higher $BI$ values, showing on average bigger lake volumes than the other units for the same landslide volumes. There is also no obvious glacial control on the stability of landslide dams (Supplementary Fig. D1). There seem to be a higher concentration of unstable landslide dams in the fluvial domain ($BI$, $DBI$, $I_s$ and $HDSI$).

## 4 Discussion

We simulated the formation of 1057 landslide dams and lakes in Austria. In the following, we discuss possible controls on the distribution of modeled dams and lakes and evaluate similarities with and differences to field observations. Finally, we provide information on model limitations.

### 4.1 Correlations of dam and lake metrics

Modeled dam and lake volumes show similar but stronger relationships than those derived from inventories, and exhibit an extended value range not observed in the field (Fig. 5). We find a clear correlation between landslide deposit volumes and dammed lake volumes in our dataset, with a Spearman-$\rho$ of 0.72. Landslide dam height proxies and landslide dam and lake volumes show similarly high correlations. In contrast, Korup (2004) reports a weaker correlation between landslide dam volumes and dammed lake volumes in New Zealand, indicated by a Spearman-$\rho$ of 0.558, and in the landslide dam datasets of Costa and Schuster (1991), Perrin and Hancox (1992) and Hancox et al. (1997). In any case, the range of our model results in almost exactly parallels uniform $II$ values (Fig. 5a), which indicates that a universal dependence of lake volumes on deposit volumes exists both in our model and in the real world.

For a given landslide volume, modeled lake volumes exhibit a bigger variability than reported in the literature (Fig. 5a). In our model, large landslides often impound relatively small lakes, leading to volume ratios ($V_{dep}/V_{lake}$) up to one order of magnitude larger than in inventories, in conjunction with low $\frac{H_{lake}}{H_{dep}}$ ratios. We relate this variability to the position of the landslide deposit in the valley. Landslides not reaching the main stream or depositing on the valley flank may only produce small lakes, and hence present a low $\frac{H_{lake}}{H_{dep}}$, while landslides depositing homogeneously across the river bed dam larger lakes and have a higher $\frac{H_{lake}}{H_{dep}}$ ratio. In contrast to our model, inventories predominantly report efficient damming in main valleys (i.e. valleys with distinct valley bottom and two flanks), while small lakes dammed by large landslides outside of clear valley structures (e.g. on valley flanks) are missed.

The negative correlation of lake volume with channel slope (Fig. 5b) can be expected as larger lakes form in higher-order sections of the drainage network where channel slopes are lower.

Modeled deposit (resp. lake) height decreases with increasing volume for large landslides, as found by Larsen et al. (2010), while small modeled landslides display an opposite scaling. We observe that $H_{dep} \sim V_{dep}{}^{0.40}$ and $H_{lake} \sim V_{dep}{}^{0.46}$ (Fig. 5c, black lines). As the exponent is greater than $\frac{1}{3}$ in both relations, the deposits become relatively thicker and the lakes become relatively deeper with increasing landslide volume. In the real world, landslide deposits reportedly show the opposite behavior. Larsen et al. (2010) obtained $V_{landslide} \sim A^{1.40}$ for both the scar area and the deposit area, which implies $H_{landslide} \sim A^{0.4}$ for the mean thickness. This thus gives $H_{landslide} \sim V_{landslide}{}^{(0.4/1.4)} = V_{landslide}{}^{0.29}$, with the depth-volume scaling exponent lower than $\frac{1}{3}$, implying that large deposits are relatively thinner than small deposits. However, thickening of deposits and deepening of lakes with increasing landslide volumes is obtained when a power-law is fitted to all model data. For the largest lake depths and dam heights relative to the deposit volumes, i.e. efficient damming, our model results mirror the inventories 5c). In contrast, thickening and deepening in our model is even more pronounced for the deposits and lakes with the smallest

heights and depths. Consequently, the power-law relationship between $V_{dep}$ and $H_{dep}$ depends on $V_{dep}$. Landslides of volumes $> 10^6$ m$^3$ show a power-law exponent of $0.448$, while landslides with volumes $> 10^7$ m$^3$ give a power-law exponent of $0.325$ (Fig. 5c). A similar relation can be observed between the lake depths and volumes (Supplementary Fig. B1). This again indicates a change in deposit geometry with $V_{dep}$ controlling the link between $V_{dep}$ and $H_{dep}$, which, upon constant model rheology, can only be attributed to valley shape.

### 4.2 Impact of glacial imprint on simulated landsliding and dam formation

Glacially-imprinted terrain hosts larger landslide and lake volumes, but lower $\frac{H_{lake}}{H_{dep}}$ ratios. This can be explained by the typical glacial topography. Glacial landscapes are characterized by overdeepened, U-shaped troughs with steep flanks, cirques, and steep arêtes and ridges that have often higher slopes than fluvial headwaters and hillslopes (Agassiz and Bettannier, 1840; Penck, 1905; Anderson et al., 2006). The formerly glaciated areas of the Austrian Alps present highest mean elevations, relief, slopes and uplift rates, and almost all modeled landslides, which also applies to the inventory (Fig. 3). Further, adjustment of glacial landscapes to deglaciation has been suggested to lead to an increase in hillslope processes (Church and Ryder, 1972; Crest et al., 2017; Jiao et al., 2018). This fits our distribution of landslides and release volumes. The landslides in glacial terrain are 2.8 times more voluminous, dam 2.5 times bigger lakes, but lead to 1.5 times lower $\frac{H_{lake}}{H_{dep}}$ ratios. We again attribute these differences to valley shape. The wide valley floors in glaciated areas demand for higher landslide volumes to dam the entire valley. Thus partial damming is more common, which leads to lower height ratios. On average, the much higher release volumes in glacial landscapes almost compensate the wide valley floors, which results in only slightly lower height ratios. This in conjunction with flat and wide valley floors leads to the formation of bigger but shallower lakes.

### 4.3 Most efficient simulated lake damming in Austria

In our model, most efficient damming, i.e. dammed lakes with exceptionally large volumes relative to the deposit volumes, occurs in several tectonic units across Austria, all characterized by exceptional valley relief. We highlight three examples found in different structural units: Gosau group, Helvetic nappes, and Tirolian nappes (Fig 4, red dots). In our simulations, large lakes are formed by landslides damming relatively narrow valleys downstream of wider and flatter valley sections. In the Gosau group, a landslide of $6.6 \times 10^6$ m$^3$ dams the Gosaubach downstream of the flat and wide Gosau valley, where a lake of $3.4 \times 10^7$ m$^3$ forms (height ratio = 0.73). In the Helvetic nappes, a landslide of $4.3 \times 10^7$ m$^3$ dams the Bregenzer Ache, leading to a lake of $5.7 \times 10^7$ m$^3$ (height ratio = 0.65). A region prone to several big landslide-induced lakes in our simulations is the Gesäuse range, which is located in the Northern Calcareous Alps. This area combines very steep valley flanks with a narrow valley floor. Consequently, the region generally presents relatively high height ratios mostly ranging from 0.38 to 0.94. The largest lake reaches a volume of $3.9 \times 10^7$ m$^3$ (height ratio = 0.56) due to valley widening upstream of the dammed gorge section of the Enns river (landslide dam volume = $5.9 \times 10^7$ m$^3$). In the same area, another landslide of $2.4 \times 10^7$ m$^3$ creates two lakes totaling $7.9 \times 10^7$ m$^3$ on the Erzbach (height ratio = 0.94). These three examples highlight the role of valley geometry in controlling the efficiency of damming. Further, our examples suggest that a change of tectonic units along a river, with a

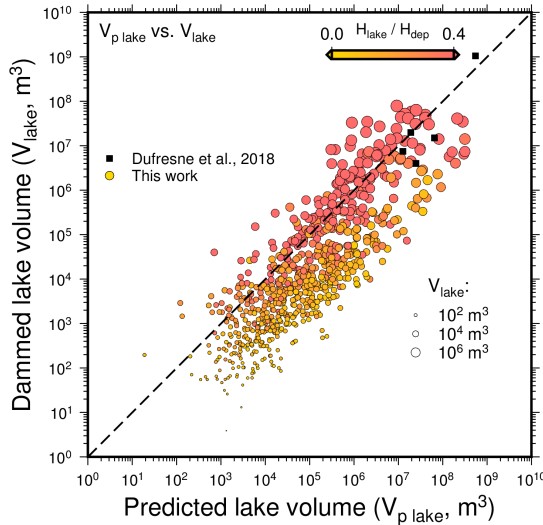

**Figure 7.** Bi-logarithmic diagram showing predicted ($V_{p\ lake}$, Eq. 7) vs. modeled ($V_{lake}$) landslide-dammed lake volume. Circle size represents dammed lake volume, circle color indicates height ratio. 1:1 relation depicted by dashed line.

narrow section at the damming location and a wider section upstream, favors efficient damming and the formation of very large lakes. In the Austrian Alps such settings occur in the Northern Calcareous Alps (e.g. Enns river, Salzach river).

### 4.4 Predicting the volume of landslide-dammed lakes

In our model results, we find a relationship between $V_{dep}$ ($= V_{landslide}$) and $V_{lake}$ (Fig. 5a), but also between $V_{lake}$ and upstream drainage area $A_b$ at the location of damming, which we use to compute a predicted lake volume $V_{p\ lake}$, such that

$$V_{lake} \sim V_{p\ lake} = \alpha \cdot V_{landslide}^{0.98} \cdot A_b^{0.92} \times 10^{-6} \tag{7}$$

with $\alpha = 0.003$ and $A_b$ in m$^2$.

The existence of such a relationship can be theoretically explained by the influence of the drainage system on valley morphology. The volume of the lake depends on the volume of the landslide and the valley shape. The width, depth (and hence height of the valley flanks) and the longitudinal slope of the valley depend on the upstream drainage area (Flint, 1974; Whitbread et al., 2015), as does the height of the dam for a given landslide volume. The relationship also applies to real world data and allows the prediction of potential $V_{lake}$ only from $V_{landslide}$ and $A_b$ (Fig. 7), two metrics that can be easily obtained from DEMs and landslide inventories. Further, the relationship facilitates the development of damming scenarios with little effort by computing potential lake volumes from different potential landslide volumes. The model explains a larger part of the variation in $V_{lake}$ ($R^2 = 0.687$) than $V_{dep}$ or $A_b$ alone (respectively $R^2 = 0.497$ and $R^2 = 0.394$). Further, the model can be approximated reasonably well by assuming a linear influence of $V_{landslide}$ and $A_b$. The additional variation of $V_{lake}$ present in the data again

depends on valley and hence deposit geometry, as indicated by the color-coded $\frac{H_{lake}}{H_{dep}}$ ratio in (Fig. 7). The prediction works best for efficient damming indicated by high $\frac{H_{lake}}{H_{dep}}$.

## 4.5 Obstruction and stability indices

The obstruction and stability indices calculated from our 1057 simulated landslide dams do not provide consistent assessments. This finding corroborates the results of Dufresne et al. (2018), who also found the indices $BI$, $II$, $DBI$, $Is$, $Ia$ and $HDSI$

inconclusive in the Eastern Alps.

However, since our model cannot directly predict the stability of the modeled landslide dams, we can only conclude that they are inconsistent but cannot rate the performance of the indices in the Austrian Alps. The $II$ and $DBI$ are the two only indices showing a relationship with the metrics of our modeled landslides, represented by $\frac{H_{lake}}{H_{dep}}$ in Fig. 6. For these indices, stability decreases with increasing $\frac{H_{lake}}{H_{dep}}$, as well as increasing catchment area, lake volume and depth. All other investigated

indices seem to depend on regionally constrained stability classes and are thus not easily transferable to the Austrian Alps.

## 4.6 Limits and amelioration of the method

### 4.6.1 Differences between simulations and inventories

Part of the discrepancies between modeled and real-world metrics (e.g. landslide and lake volume) are likely explained by topographic differences between our study area (Austrian Alps) and other mountain ranges we used for comparison. Variations

in the topographic expression are related to lithological heterogeneity (contrasts in rock mass strength), climatic conditioning (e.g. fluvial versus glacial, rates of precipitation) and tectonic forcing (variations in timing and rates of uplift). However, the differences between modeled and real-world metrics may also be a consequence of uncertainties in field measurements and oversimplifications in our model.

The accuracy of field data is limited by, among other effects, measurement uncertainties and systematic under-representation

of small landslide dams. In many cases, remnants of landslide dams and lakes need to be interpreted, hampering the assessment of their size and extent. In addition, even if dams and lakes are preserved, the topography prior to landsliding often remains unknown. This effect is also mentioned by Korup (2004), who suggests that uncertainties in the estimation of landslide dam heights are responsible for the differences between field and model results. Furthermore, large landslides may only create small dams and shallow lakes, for example when they partially block the valley floor or impound a small creek in relatively steep

terrain. Since small dams get eroded in a short time and shallow ponds of water fill with sediments very quickly, they often remain undiscovered in the field. Yet they can be simulated, leading to a wider range of modeled landslide dams. These small dams are not considered in the inventories of Fan et al. (2020), Dufresne et al. (2018), Korup (2004) and Costa and Schuster (1988). The typical dammed lake size raising interest beyond the landslide itself seems to differ between massifs. In the case of the Alps, dams are reported for $II < 2$ (Fig. 5a).

In contrast to field measurements, geomorphometric parameters obtained in a modeling study are highly precise, but assumptions and approximations made along the numerical process chain introduce uncertainty to the results. As an example, we

assume that lakes are filled to the brim, which might not always happen in reality, due to loss of water via groundwater flow through the landslide deposits or river bed substrate (Snyder and Brownell, 1996).

### 4.6.2    Uniform slope stability threshold

The determination of landslide release areas is crucial for our study. We employ an empirical model (Hergarten, 2012) that relies on the assumption of spatially uniform slope stability thresholds. We use the same slope stability thresholds for the entire Austrian Alps, which represents a distinct simplification. The study area hosts rocks that form differently steep landscapes, are characterized by potentially different rock mass strengths and therefore are likely to resist differently to erosive surface processes. It is generally assumed that rock mass strength exerts some control on slope stability thresholds on bedrock slopes

(Montgomery, 2001), which host the landslide release areas of the study region. However, this assumption has rarely been tested (Goudie, 2016) and can hardly explain the persistence of "over-steepened" valley flanks (Fernández et al., 2008) abundantly observed in glacially imprinted mid-latitude mountain ranges such as the Austrian Alps. In addition to rock type, a variety of other parameters, including weathering, tectonic stresses, type and orientation of discontinuities at different scales, influence rock mass strength (Augustinus, 1995).

However, this study focuses on regional patterns of landslide dams and lakes, and to our knowledge, no stability thresholds based on lithology or rock mass strength are available at this scale. Moreover, the model used here to determine landslide release areas (Hergarten, 2012) is so far the only model which is able to reproduce the typical power-law scaling of landslides (Supplementary Fig. E1; Tebbens, 2020). This scaling is not altered much by shifting the stability thresholds within a realistic slope range where rapid mass movements originate in mountainous areas (Hergarten, 2012). Furthermore, the power-law

scaling applies to rockfalls but also to slides (Brunetti et al., 2009). As an advantage, taking the same thresholds for the whole mountain range allows for a simple model, where topography is the main control of landsliding. Indeed, the similarities between our results (Fig. 3) and inventory events imply that topography is indeed the main control on the spatial distribution and scaling of landslides and landslide-dammed lakes on this large scale of analysis.

### 4.6.3    Lack of temporal constraints

While the employed landslide release area model (Hergarten, 2012) can provide release areas and related volumes, which cluster in the same regions as the events recorded in landslide inventories, and which are consistent with power-law scaling of landslides observed in nature, the model cannot predict timing or probability of failure of individual events. While such information would be of great value for natural hazard mitigation, neither field data as input parameters nor any of the existing state-of-the-art models can currently provide such an information at the scale of an entire mountain range. Hence, modeling

results cannot be interpreted in terms of landslide-damming probability, nor in terms of return periods, which is also far beyond the scope of this study. As a consequence, we use the term landslide "densities" for the number of landslides per area to avoid misinterpretations in terms of time dependence (e.g. probability of occurrence or recurrence interval).

### 4.6.4 Rheological model

The determination of the rheology of the moving landslide mass is crucial as the chosen flow resistance law (i.e. Voellmy rheology) and the applied parameters control the run-out distance and the landslide dam geometry (Hungr, 2011). Landslide rheology may be controlled by lithology, but may also vary spatially within a single landslide event, when different rock types are involved, or temporally, when a change in physical conditions (e.g. water content, path material) happens during the landslide runout (Hungr and Evans, 2004; Aaron and McDougall, 2019). For individual landslides, rheology parameters are in general determined by a back analysis of the event itself or events in the same region (Mergili et al., 2020). However, considering this level of detail for an entire mountain range would require back-analyzing a large number of landsliding events, which is far beyond the capabilities of this investigation.

Runout simulations are type-specific (Hungr et al., 2001; Dorren, 2003), but most of the rockfalls with $V > 10^5$ m$^3$ have a long runout (i.e termed "rock avalanche") and can be simulated accurately if the correct rheology model is used (Körner, 1976). Here, we apply the Voellmy flow resistance law with the parameter set determined by a back analysis of the well documented Val Pola landslide (Sanne, 2015) to all simulated landslides of this study. As a benefit of a uniform parameter set, we can directly compare dam geometries and related lakes across the Austrian Alps and attribute spatial variations to topography. To explore the influence of the two Voellmy parameters $\xi$ and $\mu$ on dam height, we performed a parameter study starting with the $\xi$ / $\mu$ parameter set originally determined by Sanne (2015) (Supplementary Fig. A1). The parameter study at ten different locations shows that dam height increases with $\mu$. While increasing $\xi$ causes an increase in landslide velocity and runout distance, we only observe a slight negative impact on dam height. As long as the parameter sets are suitable to describe the behavior of large landslides in alpine regions (and not mudflows or lahars with a completely different rheology unsuitable to form major dams) our parameter study implies that different rheologies will change the dam geometry to some extent but will not necessarily lead to a statistically consistent change in lake depth and volume (Supplementary Fig. A1).

### 5 Conclusions

We modeled landslides, landslide dams and dammed lakes in Austria with a new approach that combines a probabilistic approach to determine landslide release areas and a fluid dynamic model to compute landslide runouts. Based on our results, we explored relationships between properties of landslides, landslide dams and lakes, and the drainage area and valley shape.

- The resulting landslides predominantly occur in steep alpine terrain and spatially coincide with historical events reported in inventories.

- Valley geometry and the drainage system control the efficiency of damming, i.e. small landslide dams impounding large lakes. Consequently, dam and lake metrics differ for glacial and fluvial terrain.

- The modeled range in damming efficiency is much larger than in inventories, where mostly events of efficient damming are reported. In our study, scaling of landslide, dam and lake metrics differs for low and high damming efficiency.

- We provide a new relationship to estimate lake volume only from upstream drainage area and landslide volume. These two parameters explain more than 60% of lake volume variability.

- Common stability and obstruction indices do not provide concise information on dam persistence. While the Impoundment Index $II$ and the Dimensionless Blockage Index $DBI$ seem to work relatively well, the other tested indices give inconsistent results, with stability classes strongly varying between regions.

Our modeling results suggest that events with a low damming efficiency are much more frequent than represented in inventories and that they may exhibit a different scaling of landslide and lake metrics. We suspect that such events are also common in the real world and high-efficiency events are over-represented in inventories. We thus suggest that a focus is put on low-efficiency damming in the compilation of future landslide databases.

From a hazard point of view, our study statistically models the initial steps of a natural hazard cascade. A logical extension of this work to be covered in future research would thus be a dam-breaching model (Fan et al., 2019) to simulate the longevity and stability, as well as the failure mode of the created dams.

*Code and data availability.* The code is available online, and has been encapsulated in a Docker container for easy setup: DOI: 10.5281/zenodo.4171597 .

*Author contributions.* Funding acquisition: G.P. and D.H.; Conceptualization: J.R., A.-L.A. and G.P.; Methodology: A.-L.A. and J.R.; Validation: A.-L.A., J.R. and G.P.; Formal analysis: A.-L.A.; Investigation: A.-L.A.; Data curation: A.-L.A. and J.R.; Writing—original draft preparation: A.-L.A., G.P. and J.R.; Writing—review and editing: A.-L.A., G.P., J.R., S.H., L.A., D.H. and Z.D.; Visualization: A.-L.A.; Supervision: J.R. and G.P.; Project administration: G.P. and D.H.. All authors have read and agreed to the published version of the manuscript.

*Competing interests.* The authors declare that they have no conflict of interest.

*Acknowledgements.* This research has been supported by the Austrian Academy of Sciences (ÖAW) through the project RiCoLa (Detection and analysis of landslide-induced river course changes and lake formation). The authors would like to thank Franz Neubauer for the discussions on the geology of the Alps. All maps are created using the Generic Mapping Tools (Wessel et al., 2019).

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

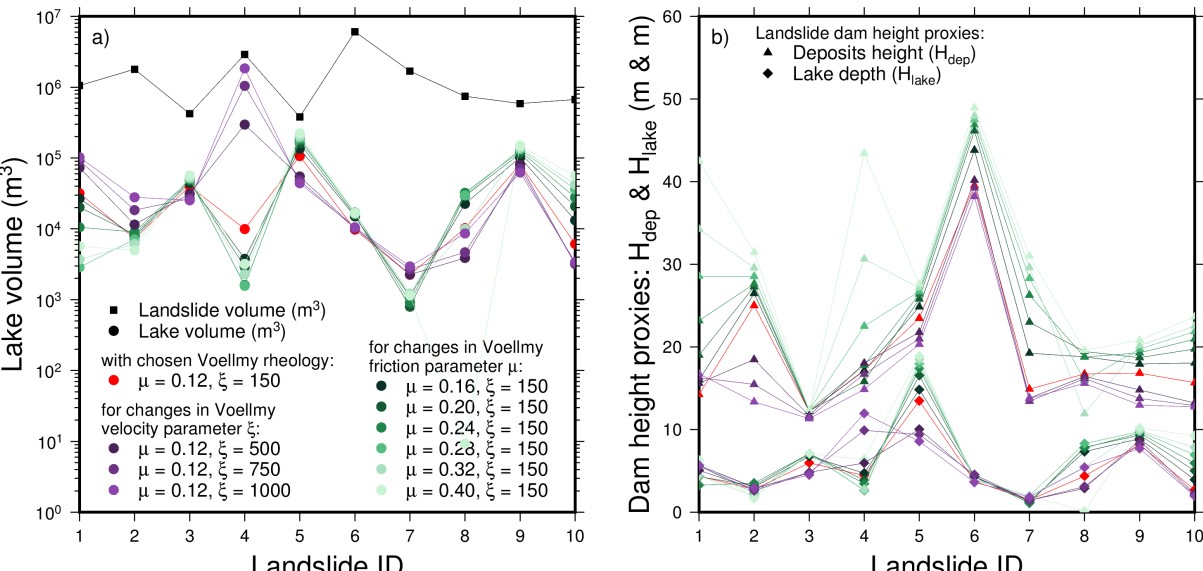

**Figure A1.** Impact of the Voellmy rheological parameters on lake volumes and landslide damming height proxies for 10 simulated landslides. The indices chosen in the simulation ($\mu = 0.12$ and $\xi = 150$) are plotted in red.

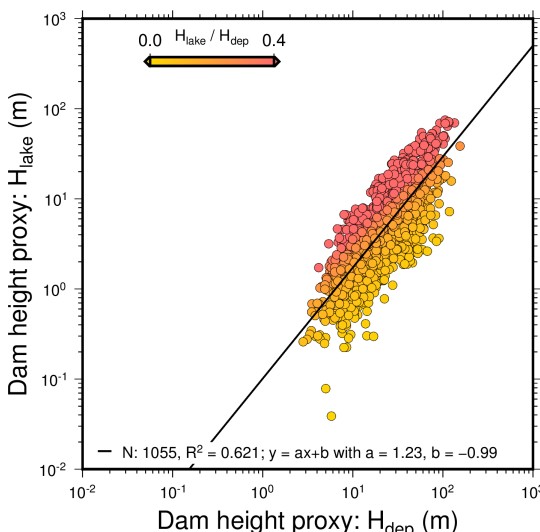

**Figure B1.** Bi-logarithmic diagram of the landslide dam height proxies: maximum lake depth $H_{lake}$ in relation to maximum landslide deposit height $H_{dep}$. We used a color gradient to highlight the change in $\frac{H_{lake}}{H_{dep}}$ ratio. We fitted power laws using least squares with vertical misfit, and indicated their sample number $N$, coefficient of determination ($R^2$) and characteristics (slope $a$ and intercept $b$).

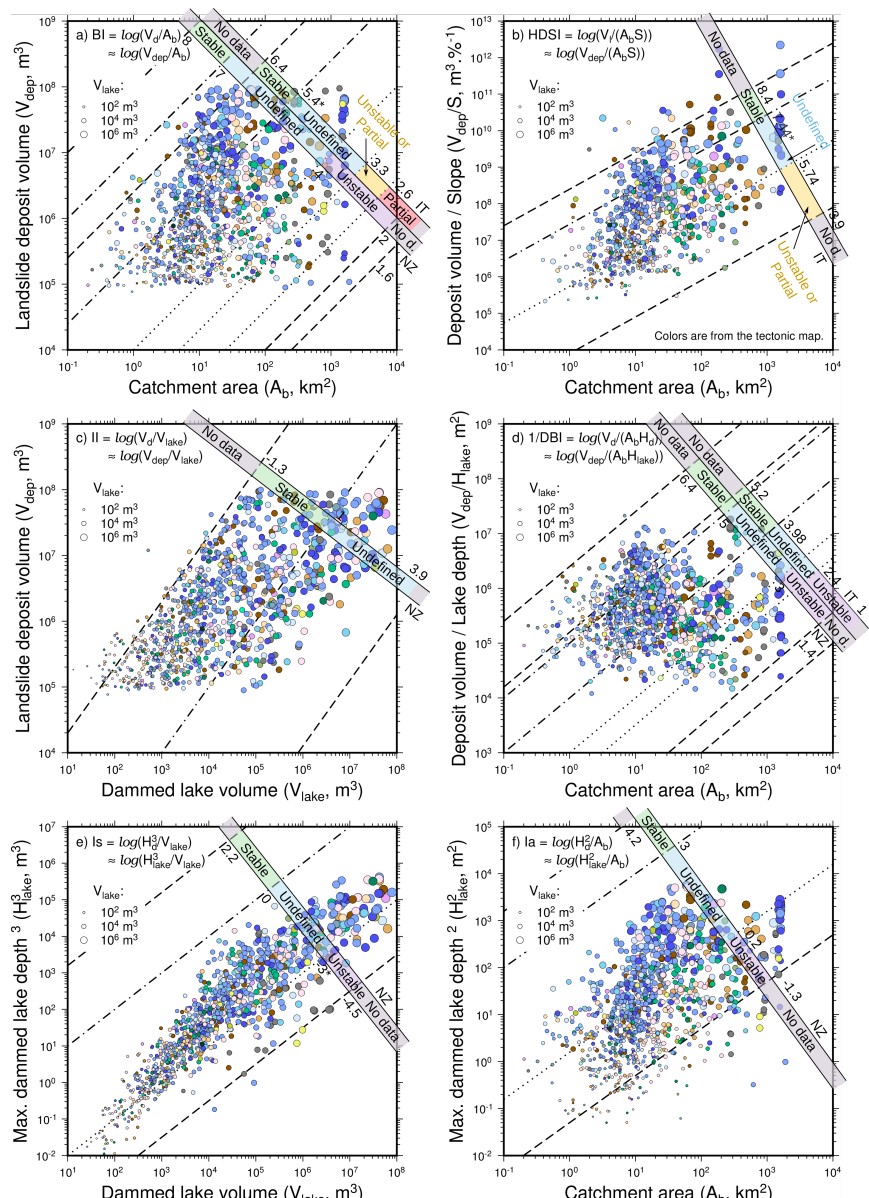

**Figure C1.** Bi-logarithmic diagrams of landslide dam classification according to two obstruction and stability indices, (a) the Blockage Index $BI$ and (b) the Hydromorphological Dam Stability Index $HDSI$, and to four stability indices, (c) the Impoundment Index $II$, (d) the Dimensionless Blockage Index $DBI$, (e) the Backstow Index $Is$ and (f) the Basin Index $Ia$. The circle color represents the tectonic unit and the circle size the logarithm of dammed lake volume. The obstruction and stability ranges from literature are indicated by scales, with the threshold values annotated on the side. Threshold lines are dashed for "No Data", dot-dashed for "Stable", dotted for "Unstable". We abbreviate NZ for New Zealand (Korup, 2004) and IT for Apennines, Italy (Tacconi Stefanelli et al., 2016). The threshold values with * present a few outliers. The cluster of values with a catchment area of $10^3$ km$^2$ are located in the same area in the Gesäuse mountain range, in the Enns catchment.

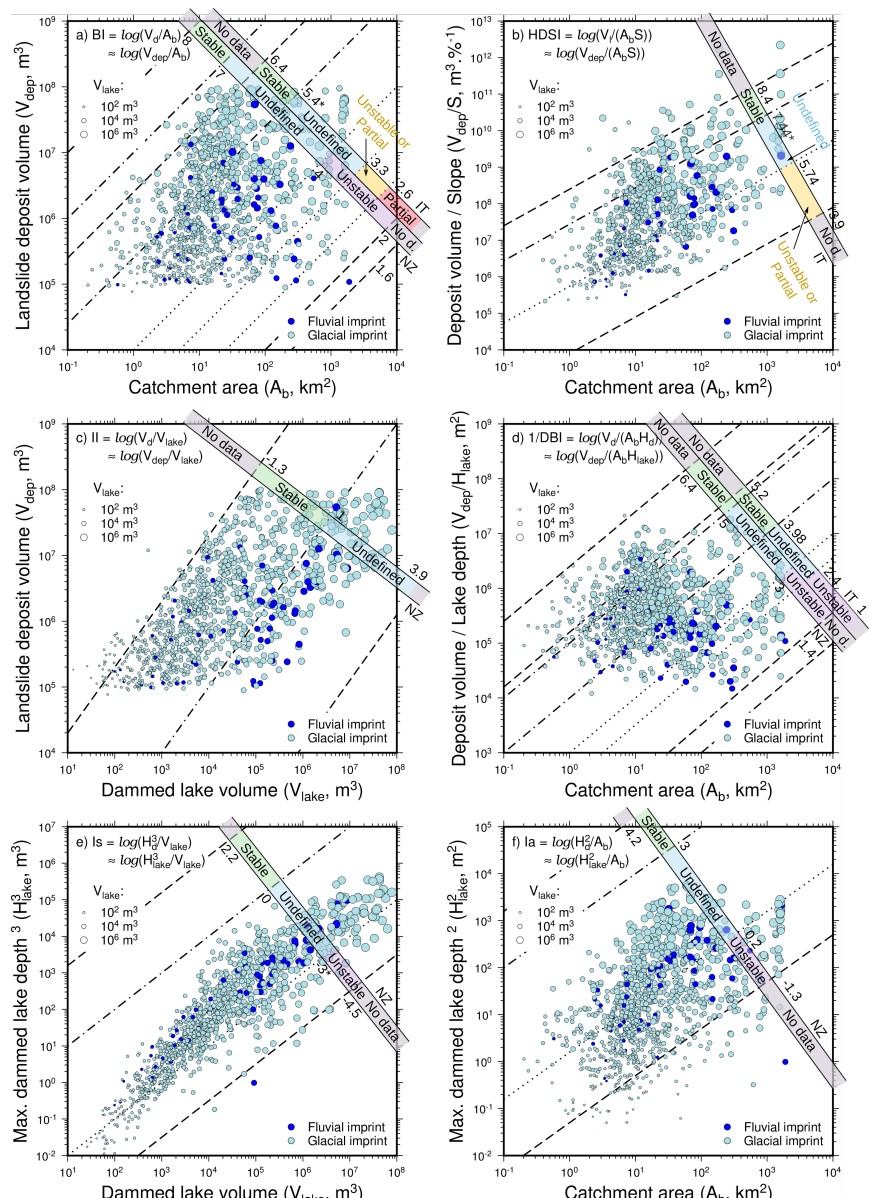

**Figure D1.** Bi-logarithmic diagrams of landslide dam classification according to two obstruction and stability indices, (a) the Blockage Index $BI$ and (b) the Hydromorphological Dam Stability Index $HDSI$, and to four stability indices, (c) the Impoundment Index $II$, (d) the Dimensionless Blockage Index $DBI$, (e) the Backstow Index $Is$ and (f) the Basin Index $Ia$. The circle color represents the glacial imprint and the circle size the logarithm of dammed lake volume. The obstruction and stability ranges from literature are indicated by scales, with the threshold values annotated on the side. Threshold lines are dashed for "No Data", dot-dashed for "Stable", dotted for "Unstable". We abbreviate NZ for New Zealand (Korup, 2004) and IT for Apennines, Italy (Tacconi Stefanelli et al., 2016). The threshold values with * present a few outliers. The cluster of values with a catchment area of $10^3$ km$^2$ are located in the same area in the Gesäuse mountain range, in the Enns catchment.

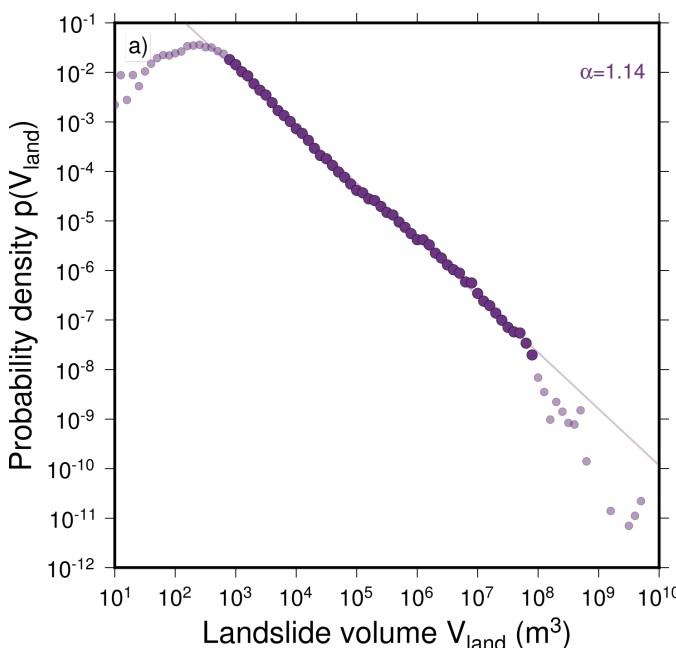

**Figure E1.** Size distribution of the landslide release volumes.