# Peer review of "Controls on the formation and size of potential landslide dams and dammed lakes in the Austrian Alps"

_Natural Hazards and Earth System Sciences, 2020_

## Referee Comment (RC1) · Anonymous Referee #1 · 23 Feb 2021

Anne-Laure Argentin et al. entitled "Controls on the formation of potential landslide dams and dammed lakes in the Austrian Alps", present a process-based modelling approach to envision susceptibility of landslide damming and lake formation by individually simulating the process chain from the initiation probability of landslides, landslide runout, river obstruction and damming. The concept and the methods employed by the authors are thought-provoking and progressive and this manuscript would be significant for the engineering geological and natural hazard community. The idea to conceptualise the landslide dam hazard chain through a process-based modelling is appreciable. Nevertheless, the study may still need some additional elements or factors that can be considered, warranting a minor to moderate revisions to the manuscript

to be accepted and this reviewer see the following suggestions shall be followed. Major specific comments: 1. Modelling the landslide release areas: The authors adopted a slope-based criterion following Hergarten et al. 2012 to determine the probability of landslide release areas. The authors do mention the reason for their choice "The approach proposed by Hergarten (2012) still seems to be the only model in this context which is able to predict the observed power-law distribution of rockfall and rockslide volumes". However, the performance of the model in a terrain with lithological variations need to be questioned. Different rocks would have different thresholds with regards to slope angle and stability. In addition, rockfalls possess strong sensitivity towards discontinuities. I would request the authors to perform a validation of their analysis of landslide probability. Is it possible to compare the landslide probability estimated by the Hergarten et al. 2012 to actual events of landslides within different geological units of the study area? It would be nice to see the performance of the model for past cases at first and then use it to predict the future. In addition, the overall work, stressing on the importance of the chain of hazards from landslide occurrence, runout, damming and lake formation seems a bit incomplete. The authors do quantify the probability of failure of each potentially unstable rock mass but, not provide a probabilistic assessment of the conditions that might trigger such instabilities (e.g., a return period of a triggering rainfall, a return period of a triggering earthquake). I suggest the authors to refer Fan et al. (2019) and add a line of discussion regarding the limitations of the landslide simulations and the validity of the assumption adopted in this study. 2. Landslide runout simulation: The authors adopted Voellmy rheology to model the landslide runout with variables $\acute{E}\dot{Z}$ = 150 m.s-2 and $\mu$ = 0.12. It is common to use such constant values for different lithologies within a large area of a numerical model. However, the same need to be justified. In general, these values are obtained through back calculation of landslide runouts using known case examples. Regarding the calibration of the models, and in particular of Gerris, the authors need to discuss the choice of their parameters. Also, the authors should discuss why they think the parameters should be the same for all the subsequent events all over the study area (e.g., why should the acceleration

remain the same? and the friction?). 3. Estimation of landslide dam geometry: It is appreciable that the author attempted to simulate the landslide dam geometry at a larger scale. Their explanation of the calculation of landslide dam volume and geometry seems simple cutting down different realities but still acceptable considering the scale of the numerical simulation. However, the limitation of the approach used in this study need to be clearly mentioned. Please refer to Hungr (2011) for more insights on a comparative study on the use of landslide runout models to predict landslide dam geometries. 4. Landslide dam characterisation: In addition to the height ratio-based characterisation of the simulated landslide dams, is it possible to identify the type of dams according to Costa and Schuster (1988); Fan et al. (2020); Hermanns et al. (2011)? The authors do mention the type of landslide dams in lines 150 using simplified planform geometry. There are also other predominant types of landslide damming based on morphology though not specific to rockfall/rockslide formed landslide dams. I would like to see some discussion regarding the preciseness of the geomorphometric parameters identified and used in this study (Table 1). 5. Dam formation and stability indices: The authors mentioned that their model cannot predict the stability of landslide dams. It is okay that the authors predict only the occurrence of landslide dam and lake formation and not the dam-breach or breach-induced flooding. However, the most significant part of this study on a hazard point of view is also to envision the relative stability of longevity of a landslide dam in the future if such events occur. On a true sense, the dam-breach and the outburst flood caused is the most threatening hazard than the landslide and damming itself. In a similar study by Fan et al. (2019), the actual dam-breach and flooding was simulated for different scenarios and the same has been compared with different empirical stability indices. I suggest the authors to refer and add some lines of expressions. I also suggest the authors to add more lines of discussion regarding the performance of stability indices. The authors do mention BI, II, DBI, Is, Ia and HDSI are inconclusive in the Eastern Alps. This also depends on the availability of data as mentioned in a previous study by Fan et al. (2020). Minor comments: 1. The authors performed a well throughout study and I appreciate their

efforts. I feel the English language presentation need improvement though myself neither a language expert or a native English speaker. 2. Since the introduction part I felt many sentences are not connected to form a nice story. The authors shall imagine the geological processes in sequence and start from the conditions of landsliding and go on write about the events until for the formation of landslide dam and lakes. This will help the readers to understand the authors are focusing on an important large-scale geological hazard chain. References cited:

Costa, J.E. and Schuster, R.L., 1988. The formation and failure of natural dams. Geological society of America bulletin, 100(7): 1054-1068. Fan, X., Dufresne, A., Siva Subramanian, S., Strom, A., Hermanns, R., Tacconi Stefanelli, C., Hewitt, K., Yunus, A.P., Dunning, S., Capra, L., Geertsema, M., Miller, B., Casagli, N., Jansen, J.D. and Xu, Q., 2020. The formation and impact of landslide dams – State of the art. Earth-Science Reviews, 203: 103116. Fan, X., Yang, F., Siva Subramanian, S., Xu, Q., Feng, Z., Mavrouli, O., Peng, M., Ouyang, C., Jansen, J.D. and Huang, R., 2019. Prediction of a multi-hazard chain by an integrated numerical simulation approach: the Baige landslide, Jinsha River, China. Landslides. Hermanns, R.L., Folguera, A., Penna, I., Fauqué, L. and Niedermann, S., 2011. Landslide dams in the Central Andes of Argentina (northern Patagonia and the Argentine northwest), Natural and artificial rockslide dams. Springer, pp. 147-176. Hungr, O., 2011. Prospects for prediction of landslide dam geometry using empirical and dynamic models, Natural and Artificial Rockslide Dams. Springer, pp. 463-477.

---

## Referee Comment (RC2) · Anonymous Referee #2 · 8 Mar 2021

I have revised the article "Controls on the formation of potential landslide dams and dammed lakes in the Austrian Alps" submitted by Anne-Laure Argentin and co-authors. The article discusses modeled landslides, landslide dams and dammed lakes, introducing an approach that combines a probabilistic approach to determine landslide release areas and a fluid dynamic model to compute runouts. The article is interesting but requires revisions before publications. A short introduction of the study area and available data should be added in chapter 2. It's not very clear what type of failures you model (debris flow, slide). Several comments are reported throughout the text. The manuscript should be revised by an English speaking person before publication.

[Figure]

Please also note the supplement to this comment:
https://nhess.copernicus.org/preprints/nhess-2020-326/nhess-2020-326-RC2-supplement.pdf

—————————————————————

[Figure]

**Supplement:**

**Controls on the formation of potential landslide dams and dammed lakes in the Austrian Alps**

[revised manuscript text omitted]

In this study, we use a modeling approach to investigate the influence of topography and glacial imprint on the potential occurrence of landslide dams and landslide-dammed lakes in Austria, and on landslide and lake characteristics. We further calculate common landslide dam obstruction and stability indices, develop a simple approach to estimate the volume of potential landslide-dammed lakes and compare our results to real-world inventories.

**2 Materials and Methods**

We use a novel combination of different numerical algorithms to model the formation of landslide dams and lakes. Our modeling workflow consists of three main steps: determination of landslide release areas and volumes, simulation of landslides, computation of geomorphometric parameters of landslide dams. Finally, we use the retrieved information to characterize and discuss dam and lake formation (Fig. 1).

**2.1 Topographical, glacial and geological datasets**

We use a freely available LiDAR-based digital elevation model (DEM) of the Austrian Alps (Open Data Österreich, starting 2015) with a spatial resolution of 10 m. The geophysical relief is based on the ASTER GDEM V3 (NASA/METI/AIST/Japan Spacesystems, and U.S./Japan ASTER Science Team, 2019). We consider the glacially overprinted terrains to be found within the mapped extent of the last glacial maximum (LGM) originating from Ehlers and Gibbard (2004). We display the mapped tectonic units of the Alps (Bousquet et al., 2012, Fig. 4) over the study area. However, as the geological and structural variability remains high within the tectonic units, we do not venture to classify them according to resistance to erosion.

[revised manuscript text omitted]
 suggest, that this can be attributed to the influence of valley geometry, such that efficient damming in well-developed valleys (i.e. valleys with distinct valley flanks) is predominantly reported in inventories, while small lakes dammed by large landslides outside of clear valley structures are missed. We further impute this variability in our results to the disposition of the deposited mass in the valley. Landslides that do

315    not reach the main stream or deposit on the valley flank may only produce small lakes and hence present a low $\frac{H_{lake}}{H_{dep}}$. On the other hand, landslides depositing homogeneously across the river bed should dam larger lakes and have a higher $\frac{H_{lake}}{H_{dep}}$ ratio, in particular in narrow valleys.

[revised manuscript text omitted]

**4.5 Obstruction and stability indices**

Our model cannot directly predict the stability of the modeled landslide dams, but we calculated several common stability and obstruction indices for our results. The obtained obstruction and stability patterns differ tremendously. A correspondence with the metrics of our modeled landslides, represented by $\frac{H_{lake}}{H_{dep}}$ in Fig. 6, is only obvious for the $II$ and the $DBI$. For these indices, stability decreases with increasing size and depth of lakes and increasing lake depth relative to deposit height. All other investigated indices seem to depend on regionally constrained stability classes and are thus not easily transferable to other regions. This finding is backed by the results of Dufresne et al. (2018), who found the $BI$, $II$, $DBI$, $Is$, $Ia$ and $HDSI$ inconclusive in the Eastern Alps.

**4.6 Limits and amelioration of the method**

Topographic and other differences between mountain ranges likely explain part of the differences between modeled and real-world metrics and correlations, but they may also be a consequence of uncertainties in field measurements and oversimplifications in the models.

The accuracy of field data is, among other effects, limited by measurement uncertainties and systematic under-representation of small landslide dams. In many cases, remnants of landslide dams and lakes need to be interpreted, hampering the assessment of their size and extent. If dams and lakes are preserved, the topography prior to landsliding is often unknown. Korup suggests that uncertainties in the estimation of landslide dam heights are responsible for the difference between field and model results. Furthermore, large, high deposits may often create only small, shallow lakes, for example when they only partially block the valley floor or impound a small creek in relatively steep terrain. Small landslide dams and lakes often remain undiscovered in the field. Small dams may either only exist for a short time or shallow ponds of water may fill with sediments very quickly. Thus, they can hardly be accounted for in field surveys, while they can be simulated, leading to a wider range of modeled landslide dams. These small dams are not considered in the inventories of Dufresne et al. (2018), Korup (2004) and Costa and Schuster (1988). Depending on the massif, the typical range where the dam receives interest beyond the landslide is different. In the case of the Alps, this range is $II < 2$ (Fig. 5a).

Simulations, however, tend to oversimplify reality and are based on various assumptions. We introduce simplifications in determining landslide release volumes and modeling fluid flow. These assumptions influence the shape and size of the deposits and their location relative to the river bed, which further controls the amount of impounded water. However, we use approaches and spatially uniform parameters validated in other studies (Hergarten, 2012; Sanne, 2015; Hergarten and Robl, 2015). Further, we assume that lakes are filled to the brim, which might not always happen in reality, due to loss of water via groundwater flow through the landslide deposits or river bed substrate (Snyder and Brownell, 1996).

In our model to determine landslide release areas, we applied uniform stability thresholds, which are generally not well constrained and may also differ for different rock types. Thus, our model may not be able to reproduce the spatial distribution of landsliding. However, landslide inventories indicate that this is not the case for large, rapid mass movements on which we focus in this study, as large rock avalanches predominantly occur in steep landscapes with excessive relief made of strong rocks (Fig. 3). We thus conclude that our approach is suitable to qualitatively reproduce the distribution of potential large landslides and impounded lakes in a steep mountain range and to derive relationships between dam and lake size, the drainage system and valley morphology.

**5 Conclusions**

We modeled landslides, landslide dams and dammed lakes in Austria with a new approach that combines a probabilistic approach to determine landslide release areas and a fluid dynamic model to compute landslide runouts. Based on our results, we explored relationships between properties of landslides, landslide dams and lakes, and the drainage system and valley shape.

- The resulting landslides predominantly occur in steep alpine terrain and spatially coincide with historical events reported in inventories.

- Valley geometry and the drainage system control the efficiency of damming, i.e. small landslide dams impounding large lakes. Consequently, dam and lake metrics differ for glacial and fluvial terrain.

- The modeled range in damming efficiency is much larger than in inventories, where mostly events of efficient damming are reported. In our study, scaling of landslide, dam and lake metrics differs for low and high damming efficiency.

- We provide a new relationship to estimate lake volume only from upstream drainage area and landslide volume. These two parameters explain more than 60% of lake volume variability.

- Common stability and obstruction indices do not provide concise information on dam persistence. While the $II$ and the $DBI$ seem to work relatively well, the other tested indices give inconsistent results, with stability classes strongly varying between regions.

Our modeling results suggest that events with a low damming efficiency are much more frequent than represented in inventories and that they may exhibit a different scaling of landslide and lake metrics. We suspect that such events are also common in the real world and high-efficiency events are over-represented in inventories. We thus suggest that a focus is put on low-efficiency damming in the compilation of future landslide databases.

*Code and data availability.* The codes and data are available online, with the exception of the landslide release areas and volumes determination code.

[revised manuscript text omitted]

**Table 2.** Landslide dam and lake volume ranges from around the world compared to our generated landslide-dammed lakes. The Chinese landslide dams all originate from the Wenchuan earthquake. Numbers are approximates. * *Modeled landslide dams and lakes.* ** *The modeled landslides with volume below* $10^5$ m$^3$ *were not computed.* ° *The* $H_{dam}$ *proxies are written* $H_{lake}$ | $H_{dep}$. °° *Except the Tangjiashan landslide dam outlier which impounded* $3 \times 10^8$ m$^3$ *of water.*

[revised manuscript text omitted]

---

## Author Comment (AC1) · 1 Apr 2021

The comment was uploaded in the form of a supplement:
https://nhess.copernicus.org/preprints/nhess-2020-326/nhess-2020-326-AC1-supplement.pdf

---

## Author Comment (AC2) · 1 Apr 2021

**Authors' Response to Reviews of**

**"Controls on the formation of potential landslide dams and dammed lakes in the Austrian Alps"**

Anne-Laure Argentin et al.

*Natural Hazards and Earth System Sciences,* `https://doi.org/10.5194/nhess-2020-326`
* * *
RC: *Reviewers' Comment*,    AR: Authors' Response

**1. Anonymous Referee #1,**

**RC:** *Anne-Laure Argentin et al. entitled "Controls on the formation of potential landslide dams and dammed lakes in the Austrian Alps", present a process-based modeling approach to envision susceptibility of landslide damming and lake formation by individually simulating the process chain from the initiation probability of landslides, land-slide runout, river obstruction and damming. The concept and the methods employed by the authors are thought-provoking and progressive and this manuscript would be significant for the engineering geological and natural hazard community. The idea to conceptualize the landslide dam hazard chain through a process-based modeling is appreciable. Nevertheless, the study may still need some additional elements or factors that can be considered, warranting a minor to moderate revisions to the manuscript paper to be accepted and this reviewer see the following suggestions shall be followed.*

 AR: We thank the reviewer for the thorough review of our manuscript and the pertinent comments formulated. We are very glad that the reviewer finds our article a significant addition to the scientific community and are confident that we can address the issues raised in a revised version of the article. In this response we reply line by line to the suggestions made:

**1.1. Major comment #1: Modeling the landslide release areas**

**RC:** *The authors adopted a slope-based criterion following Hergarten et al. 2012 to determine the probability of landslide release areas. The authors do mention the reason for their choice "The approach proposed by Hergarten (2012) still seems to be the only model in this context which is able to predict the observed power-law distribution of rockfall and rockslide volumes". However, the performance of the model in a terrain with lithological variations need to be questioned. Different rocks would have different thresholds with regards to slope angle and stability.*

 AR: The first method used from Hergarten (2012) is an empirical method, which is not physically process-based. Although it seems intuitive that different rocks would have different thresholds with regards to slope angle and stability, as formulated for soil-mantled slopes and bedrock slopes (Montgomery, 2001), it has seldom been tested with bedrock slopes (Goudie, 2016) and does not explain some "over-steepened" slopes (Fernández et al., 2008). Furthermore, rock mass strength is a variable that is controlled by a large set of parameters (e.g. lithology, structural discontinuities etc.), and, to our knowledge, no thresholds are available based on rock mass strength for our study area.

**RC:** *In addition, rockfalls possess strong sensitivity towards discontinuities. I would request the authors to perform a validation of their analysis of landslide probability.*

**RC:** *Is it possible to compare the landslide probability estimated by the Hergarten et al. 2012 to actual events of landslides within different geological units of the study area? It would be nice to see the performance of the model for past cases at first and then use it to predict the future.*

**AR:** The reviewer is right to mention the influence of lithology and discontinuities on landslide triggering. Our method uses a statistical approach that only holds for extended regions, does not take into account lithological variations and structural discontinuities, and thus cannot be applied to reproduce case studies. Moreover, we assume that the different stability thresholds lead to an equafinality of results. Taking the same thresholds for the whole mountain range allows for a simple model. The topography is the main control of landsliding here. Furthermore, no temporal constraints are applied to our model, and we do not investigate any triggering return periods or landslide frequency. We thus call "density" and not "frequency" the number of landslide simulated per km$^2$. This density is closer to a landsliding potential, with high densities where not much landsliding has already occurred (i.e. in steep terrains).

We validated our model by visually comparing the landslides created with a landslide database over the Austrian Alps. However, to provide a better overview of the difference between landslide densities per tectonic unit, created a histogram of the landsliding frequency in each lithological unit for both our model and the database:

[Figure]

Figure 1: Comparison of landslide density depending on the Alpine tectonic unit. Each ring corresponds to a different dataset and the arcs length represents the landslide frequency for every tectonic unit.

The Tirolian nappes show a higher density of landslides in our model than in the datasets from Kuhn (visited 2020.07.27) (Fig. 1), which is logical since the time range investigated by the database is restrained. We notice that 1) the (computationally-driven) decision to choose landslides with $V > 10^5$ m$^3$ exacerbates differences with landslide database densities, 2) landslide densities are overestimated in steep terrains (e.g. calcareous nappes).

However, this model recreates the typical power-law scaling of landslides (Fig. 2, Tebbens, 2020), and changing the thresholds do not change the size distribution of landslides by much (Hergarten, 2012).

[Figure]

Figure 2: The landslide scaling of the landslides simulated in the model.

We will clarify the aforementioned points in the discussion. We will expand our paragraph to fully cover the limits of the method the reviewer signaled.

**RC:** *In addition, the overall work, stressing on the importance of the chain of hazards from landslide occurrence, runout, damming and lake formation seems a bit incomplete. The authors do quantify the probability of failure of each potentially unstable rock mass but, not provide a probabilistic assessment of the conditions that might trigger such instabilities (e.g., a return period of a triggering rainfall, a return period of a triggering earthquake). I suggest the authors to refer Fan et al. (2019) and add a line of discussion regarding the limitations of the landslide simulations and the validity of the assumption adopted in this study.*

**AR:** We will add a paragraph on the absence of a timescale in our model, which prevents us from discussing the return period of triggering events. As a result, we talk about landslide "densities" for the number of landslides per area. This will be investigated in following work.

**1.2. Major comment #2: Landslide runout simulation and #3: Estimation of landslide dam geometry**

**RC:** *The authors adopted Voellmy rheology to model the landslide runout with variables $\xi = 150m.s^{-2}$ and $\mu = 0.12$. It is common to use such constant values for different lithologies within a large area of a*

*numerical model. However, the same need to be justified. In general, these values are obtained through back calculation of landslide runouts using known case examples. Regarding the calibration of the models, and in particular of Gerris, the authors need to discuss the choice of their parameters. Also, the authors should discuss why they think the parameters should be the same for all the subsequent events all over the study area (e.g., why should the acceleration remain the same? and the friction?).*

**RC:** *It is appreciable that the author attempted to simulate the landslide dam geometry at a larger scale. Their explanation of the calculation of landslide dam volume and geometry seems simple cutting down different realities but still acceptable considering the scale of the numerical simulation. However, the limitation of the approach used in this study need to be clearly mentioned. Please refer to Hungr (2011) for more insights on a comparative study on the use of landslide runout models to predict landslide dam geometries.*

 AR: Two very good points, indeed. We chose to discuss and take into account the comments #2 and #3 together since they are strongly linked.

Rheology determination is important and especially tricky for landslide runout modeling. Rheology matters because its choice controls the landslide dam geometry (Hungr, 2011), as shown by the consistent impact our Voellmy parameters have on dam height (Supplementary Fig. A1). However, rheology determination is not an easy task, and usually needs the back analysis of a case study. Since rheology is linked to lithology, different landslides will present different runout rheologies. Rheology can also vary spatially in a single landslide event, when two different rock types are involved, or temporally, when a change in physical conditions happens during the landslide runout (Hungr and Evans, 2004).

However, we have no way of knowing how the rheology might vary over the study area, let alone temporally or spatially in a single case study. Using the same rheology over the whole Austrian Alps enables an easier analysis of the results based on the topography. If we were to add too much complexity, we would not be able to infer which effect controls which result.

Furthermore, landslide dam geometry influences the lake geometry in a complex manner. We show that although the Voellmy parameters have a consistent impact on dam height, this does not translate to consistent changes in lake depth or lake volume (Supplementary Fig. A1). Thus we assume that different rheologies would not necessarily lead to a statistically significant change in lake volumes.

For those two reasons, we decided to use the same rheological coefficients for all events in our study. We assume that all landslides are similar and exhibit the same rheology. Furthermore, using homogeneous parameters ensures we can easily compare the resulting dam geometries. Thus, we relied on the back analysis of landslide runout from Sanne (2015) on the Val Pola event to set our Voellmy rheology parameters.

We will add a paragraph on rheology and the use of landslide runouts to predict landslide dam geometries.

**1.3. Major specific comment #4**

**RC:** *Landslide dam characterization: In addition to the height ratio-based characterisation of the simulated landslide dams, is it possible to identify the type of dams according to Costa and Schuster (1988); Fan et al. (2020); Hermanns et al.(2011)? The authors do mention the type of landslide dams in lines 150 using simplified planform geometry. There are also other predominant types of landslide damming based on morphology though not specific to rockfall/rockslide formed landslide dams.*

 AR: In the current state of our work, a planform characterization of the simulated landslide dams would be difficult: it implies an automatic recognition of shapes in planform view. Our dams sometimes exhibit complex shapes, with landslide deposits spread across valley flanks and valley floors. The planform geometries would

require we make the distinction between deposits that sedimented on the valley flanks and those that actively contribute to valley damming. Furthermore, some landslides separated in two valleys during their runouts, and thus form two distinct deposit areas. This distinction would require another non-trivial algorithm. The reviewer comment is however a very good idea for further work.

RC: *I would like to see some discussion regarding the preciseness of the geomorphometric parameters identified and used in this study (Table 1).*

AR: We will add a few sentences which explains the difference between preciseness and uncertainty for our geomorphometric parameter, and thus what are its limits and how accurate it is deemed.

**1.4. Major specific comment #5**

RC: *Dam formation and stability indices: The authors mentioned that their model cannot predict the stability of landslide dams. It is okay that the authors predict only the occurrence of landslide dam and lake formation and not the dam-breach or breach-induced flooding. However, the most significant part of this study on a hazard point of view is also to envision the relative stability of longevity of a landslide dam in the future if such events occur. On a true sense, the dam-breach and the outburst flood caused is the most threatening hazard than the landslide and damming itself. In a similar study by Fan et al. (2019), the actual dam-breach and flooding was simulated for different scenarios and the same has been compared with different empirical stability indices. I suggest the authors to refer and add some lines of expressions.*

AR: We will add a few lines on the necessity to model the dam-breach and simulate the flood to assess the hazards coming from landslide-dam failures.

RC: *I also suggest the authors to add more lines of discussion regarding the performance of stability indices. The authors do mention BI,II, DBI, Is, Ia and HDSI are inconclusive in the Eastern Alps. This also depends on the availability of data as mentioned in a previous study by Fan et al. (2020).*

AR: We will reformulate this part. The reviewer's concerns about the availability of data (Fan et al., 2020) apply to the study of Dufresne et al. (2018), which found the BI, II, DBI, Is, Ia and HDSI inconclusive in the Eastern Alps based on a handful of case studies. However, our study presents a total of 1057 events, and those events do not present consistent stability assessments across indices. Although we cannot infer from this study which indice and which thresholds are best suited for the Eastern Alps, we can conclude that they do not agree with each other.

**1.5. Minor comment #1**

RC: *The authors performed a well throughout study and I appreciate their efforts. I feel the English language presentation need improvement though myself neither a language expert or a native English speaker.*

AR: We noted the reviewer comment and will ask for English feedback from colleagues.

**1.6. Minor comment #2**

RC: *Since the introduction part I felt many sentences are not connected to form a nice story. The authors shall imagine the geological processes in sequence and start from the conditions of landsliding and go on write about the events until for the formation of landslide dam and lakes. This will help the readers to understand the authors are focusing on an important large-scale geological hazard chain.*

AR: We will re-write more clearly the article and highlight its storyline by relying on transitional words and

coherence methods. We will reformulate the article based on recommendations from a science-based guide to writing.

References cited by the reviewer:

Costa, J.E. and Schuster, R.L., 1988. The formation and failure of natural dams. Geological society of America bulletin, 100(7): 1054-1068.

Fan, X., Dufresne, A., SivaSubramanian, S., Strom, A., Hermanns, R., Tacconi Stefanelli, C., Hewitt, K., Yunus,A.P., Dunning, S., Capra, L., Geertsema, M., Miller, B., Casagli, N., Jansen, J.D. and Xu, Q., 2020. The formation and impact of landslide dams – State of the art. Earth-Science Reviews, 203: 103116.

Fan, X., Yang, F., Siva Subramanian, S., Xu, Q.,Feng, Z., Mavrouli, O., Peng, M., Ouyang, C., Jansen, J.D. and Huang, R., 2019. Prediction of a multi-hazard chain by an integrated numerical simulation approach: the Baige landslide, Jinsha River, China. Landslides.

Hermanns, R.L., Folguera, A.,Penna, I., Fauqué, L. and Niedermann, S., 2011. Landslide dams in the Central Andesof Argentina (northern Patagonia and the Argentine northwest), Natural and artificial rockslide dams. Springer, pp. 147-176.

Hungr, O., 2011. Prospects for prediction of landslide dam geometry using empirical and dynamic models, Natural and Artificial Rockslide Dams. Springer, pp. 463-477.

**2. Anonymous Referee #2,**

**RC:** *I have revised the article "Controls on the formation of potential landslide dams and dammed lakes in the Austrian Alps" submitted by Anne-Laure Argentin and co-authors. The article discusses modeled landslides, landslide dams and dammed lakes, introducing an approach that combines a probabilistic approach to determine landslide release areas and a fluid dynamic model to compute runouts. The article is interesting but requires revisions before publications. A short introduction of the study area and available data should be added in chapter 2. It's not very clear what type of failures you model (debris flow, slide). Several comments are reported throughout the text. The manuscript should be revised by an English speaking person before publication.*

**AR:** We thank the reviewer for the thorough review of our manuscript and the pertinent comments formulated. We are pleased that the reviewer found our article interesting. Some of the raised issues originate from a lack of clarity on our part and we are convinced we can overcome those problems with some restructuring and a better wording. We discuss here the revisions suggested by the reviewer line by line.

**2.1. Comments**

**RC:** *l. 2: the entire territory of the Austrian Alps?*

**AR:** Yes.

**RC:** *l. 3: it's not very clear which type of landslides? & l. 4: debris flow? & l. 72: Describe better which type of landslides you are considering in your modelling. Rockfall and rockslide? Debris flow? & l. 78-79: (About slope thresholds) this is quite different from rockfall or rockslide & l. 96: (About runout simulation) Now you are not considering the failures as rock fall (line 74). Correct? & l. 114-115: (About runout simulation) You have selected "The approach proposed by Hergarten (2012) still seems to be the only model in this context which is able to predict the observed power-law distribution of rockfall and rockslide volumes" and now rockfall are not evaluated. Explain better.*

**AR:** Good questions. We do not specify the type of landslide, as this model (triggering + runout simulation) can be applied to any landslide type with high volume. The triggering model from Hergarten has been defined to reproduce the statistic distribution of rockfalls (Hergarten, 2012). Since slides also follow the same distributions (Brunetti et al., 2009) we can use the same algorithm. Runout simulations are indeed type-specific (Hungr et al., 2001), but most of rockfalls with $V > 10^5$ m$^3$ would have a long runout and be termed "rock avalanche" (Dorren, 2003). Rock avalanche runouts can be simulated accurately if the correct rheology is used (e.g. Val Pola Sanne, 2015). We chose to use the same rheology for all events to keep a simple dataset.

We will discuss this landslide type question more in detail in the discussion.

**RC:** *l. 7: "small landslides damming large lakes" is the opposite of "lake volume increases linearly with landslide volume" that you say in the same sentence & l. 9-11: what do you mean with more efficient?*

**AR:** We meant to define what we call "efficient damming", but we will reformulate the text since it is not clear.

**RC:** *l. 55: Add a short description of the study area and the available data, including the inventories you mention in the text.*

**AR:** We will add the requested information to the manuscript.

**RC:** *l. 62: which is the resoluton?*

AR: The ASTER DEM has a 1 arc second resolution. We will add this information to the article.

**RC:** *l. 65-66: this semtence here has no real meaning*

AR: We will reformulate the sentence.

**RC:** *l. 68: moving window*

AR: We will change the wording.

**RC:** *l. 73-74: About "The approach proposed by Hergarten (2012) still seems to be the only model in this context which is able to predict the observed power-law distribution of rockfall and rockslide volumes." -> Can you justify better this choice? Can you justify better the context of your application that justify this choice? & l. 88-89: (On the dependency of slope thresholds on lithology) In your analysis you have assumed that this statement it's acceptable. Is this reasonable in the test area?*

AR: Yes, we will add a paragraph at the end of the discussion to talk about the use and limitations of this method. The context of this study is the requirement to use a computationally efficient algorithm. We will reformulate this part.

Very good question. The model we use for this simulation is empirical and reproduces the size frequency distribution of landslides at the mountain range scale (Fig. 2). We answered this question in more detail in the reply to the Major comment #1 of Referee #1.

**RC:** *l. 75: explain better*

AR: We will add some explanation.

**RC:** *l. 79: landslide area or volume?*

AR: Well, it's both at the same time. The source area is expanding when the slope of the neighboring pixels is not stable, thus also increasing the landslide volume.

**RC:** *l. 83: how do you evaluate the thickness?*

AR: With the algorithm mentioned above, we remove material until a stable slope condition is reached. The height of removed material per pixel is the evaluated thickness of the landslide at the pixel location.

**RC:** *l. 86: Removed typo.*

AR: Thank you.

**RC:** *l. 92: (About memory issues) which is the extent of your study area?*

AR: The study area is the whole of Austria, and the DEM weights more than 7.4 Go. Some computations do not support such a massive input area.

**RC:** *l. 93: this means that the tiles are overlapping? what is the dimension of the buffer?*

AR: Exactly, the tiles are overlapping because we need space to simulate the landslide runouts. However, those overlapping areas are only taken once into account for triggering landslides. The buffer dimension is 10.25 km.

**RC:** *l. 107-108: It's not clear how did you model the lake volume starting from the runout of the landslides. Cases shown in fig. A1, are real cases?*

AR:   I see, I guess we mention this too early on, since we have not yet explained how we obtained the lake volumes. We will restructure the subsection.

No, the cases shown in A1 are simulated cases. We will modify the figure caption.

**RC:**   *l. 123-124: not clear*

AR:   Just a technical detail. We need to cut the DEMs again in smaller parts for the landslide simulations. We will delete this sentence which seems to distract the readers from the process.

**RC:**   *l. 161: can you explain better?*

AR:   We will reformulate the sentence.

**RC:**   *l. 162: In most of the indices you have Volumes. Explain how did you compute them*

AR:   The landslide volumes are computed using the algorithm from Hergarten as described previously: the total of all removed material on top of each pixel in the source area constitutes the landslide volume. The lake volume is computed by filling the DEM (with simulated landslide) with a common GIS algorithm and making the difference with the topography before filling. This is explained in the previous sections.

**RC:**   *l. 165 & 168: explain*

AR:   $A_b$ is the catchment area upstream of the landslide dam (l. 138 & l. 141). We will modify the text to remind the reader of the meaning of $A_b$ so they do not have to scroll back up.

**RC:**   *l. 191: 1) how did you selected the seed pixel? 2) this the procedure proposed by Hargarten. Correct?*

AR:   1) The seed pixel is randomly chosen. 2) Yes, exactly.

**RC:**   *l. 192: did you select all the Austrial Alps?*

AR:   Yes, we launched the simulation on all the Austrian Alps.

**RC:**   *l. 193: 1) Is this an available dataset? if this is the case it should be described before (in the material) 2) can you add this data and its use in the workflow ?*

AR:   No this is not an available dataset, this is the result of the simulation on the Austrian Alps. To avoid any confusion, we will add a priming sentence to the result sections, and we will modify the title of the first section.

**RC:**   *l. 197-198: not clear*

AR:   We will reformulate.

**RC:**   *l. 198-199: high slope angle?*

AR:   Yes, definitely. If the local slope is high, the landslide density will be higher. We will reformulate.

**RC:**   *l. 200: what do you mean?*

AR:   We will change the formulation.

**RC:**   *l. 202-203 & 204-205: it is very difficult to find the formations in the map becouse colours are too similar*

AR:   We will try to improve the readability of the map by numbering the tectonic units on the map and in the legend.

**RC:** *l. 206: add reference*

AR: Ok, we will add a reference.

**RC:** *l. 211: Did you mapped fluvial and glacial valleys in the entire Austria? In table 3 what is the area? How did you compute the area of density? km2 of what? glacial and fluvial valleys? I would expect larger volumes in glacial valley but smaller density.*

AR: No, we used the extent of the Last Glacial Maximum to deduce the extent of glacially imprinted landscapes. The area in Table 3 is the glacially imprinted area (glacial) and non-glacially imprinted one (fluvial). We computed the densities by dividing the number of landslides happening in the glacially imprinted landscapes (resp. fluvially imprinted) by the glacial area (resp. fluvial area). The unit is thus in "landslide.km$^{-2}$".

Yes, very good point. We indeed have larger landslide volumes in glacially imprinted areas. However, glacial valleys are prone to high landslide frequencies following glacial recession (Hartmeyer et al., 2020). Nonetheless, the "density" we are talking about in this section is not a "frequency", since we do not include any temporal constraints in our model. We will add an explanatory paragraph in the discussion about this.

**RC:** *l. 263-264: Can you add in the figure the real data as done before?*

AR: Yes, we will try to add the real data to the Figure 6 as done for the Figure 5.

**RC:** *l. 263-264: Have you tried to plot in different graphs fluvial and glacial landslide dams? & l. 291: Have you tried to plot separately glacial and fluvial data?*

AR: Yes, we plotted in Supplementary D1 the fluvial and glacial landslide dams separately.

**RC:** *l. 264: can you explain why there are many undefined?*

AR: Yes, a range was said to be "Undefined" when the events found in it exhibited variable behaviors (Partial obstruction, Complete Obstruction, Stable, Unstable). The "Undefined" ranges are particularly wide since no consistent trend was found in it (Korup, 2004; Tacconi Stefanelli et al., 2016). We will change the formulation.

**RC:** *l. 296: English is not always very clear: it should be revised by an English speaking person*

AR: We will ask one of our colleagues to revise the manuscript. We will also modify the manuscript to make the storyline easier to follow.

**RC:** *l. 311: ?*

AR: "in" We will correct our typographical error.

**RC:** *l. 313 & 314: not clear*

AR: We will restructure this paragraph.

**RC:** *l. 320: I do not see the valley geometry in the discussion below (chapter 4.1)*

AR: We are indeed not using any valley geometry metrics in this section, but we think that the valley topography partly explains the relations highlighted here. We will change the wording of this sentence.

**RC:** *l. 346: It's a little boit confusing what is is real and what has been modelled.*

AR: All this section discusses simulation results. We will add some precision to the subsection titles.

**RC:** *l. 348: Do you mean Alpine regions?*

AR:    Yes, we mean the Alpine regions, more precisely the Alpine tectonic units. We will change "regions" to "tectonic units".

RC:    *l. 371: where is the Ab in the graph?*

AR:    $A_b$ and $V_{landslide}$ are not directly plotted in the graph. We compute $V_{p\ lake}$ from $A_b$ and $V_{landslide}$ and then plot $V_{p\ lake}$ against $V_{lake}$. We will modify the graph caption and the equation and corresponding paragraph to avoid any confusion.

RC:    *l. 387: which one?*

AR:    Climate and tectonics are two of those big differences between mountain ranges. Climate and tectonics include among other parameters precipitation rates and earthquakes. Those parameters are two of the main triggers for landsliding. Precipitation rates, in particular, influence the rheology of the landslides (Chen and Lee, 2003; Wang and Sassa, 2003) and thus the geometry of the formed landslide dams (Hungr, 2011). Thus rain-triggered landslides exhibit a different shape than earthquake-triggered ones (Chen et al., 2014). We will reformulate the sentence.

RC:    *l.392-393: do you agree?*

AR:    Yes, we agree with Korup and think uncertainties in the estimation of dam heights are partly responsible for differences between field and model results. We will modify the sentence.

RC:    *l. 417: there not much evaluations on the drainage system*

AR:    We will modify "drainage system" to "drainage area".

RC:    *l. 426 & 427: write the complete name*

AR:    Ok, we will add "Impoundment Index" and "Dimensionless Blockage Index" to the abbreviations.

**References**

Brunetti, M. T., Guzzetti, F., and Rossi, M.: Probability distributions of landslide volumes, Nonlinear Processes in Geophysics, 16, 179–188, , URL `www.nonlin-processes-geophys.net/16/179/2009/`, 2009.

Chen, H. and Lee, C. F.: A dynamic model for rainfall-induced landslides on natural slopes, Geomorphology, 51, 269–288, , 2003.

Chen, K. T., Kuo, Y. S., and Shieh, C. L.: Rapid geometry analysis for earthquake-induced and rainfall-induced landslide dams in Taiwan, Journal of Mountain Science, 11, 360–370, , 2014.

Dorren, L. K.: A review of rockfall mechanics and modelling approaches, Progress in Physical Geography, 27, 69–87, , 2003.

Dufresne, A., Ostermann, M., and Preusser, F.: River-damming, late-Quaternary rockslides in the Ötz Valley region (Tyrol, Austria), Geomorphology, 310, 153–167, , URL `http://linkinghub.elsevier.com/retrieve/pii/S0169555X18301144`, 2018.

Fan, X., Dufresne, A., Siva Subramanian, S., Strom, A., Hermanns, R., Tacconi Stefanelli, C., Hewitt, K., Yunus, A. P., Dunning, S., Capra, L., Geertsema, M., Miller, B., Casagli, N., Jansen, J. D., and Xu, Q.: The formation and impact of landslide dams – State of the art, 203, , 2020.

Fernández, T., Irigaray, C., El Hamdouni, R., and Chacón, J.: Correlation between natural slope angle and rock mass strength rating in the Betic Cordillera, Granada, Spain, Bulletin of Engineering Geology and the Environment, 67, 153–164, , 2008.

Goudie, A. S.: Quantification of rock control in geomorphology, , 2016.

Hartmeyer, I., Delleske, R., Keuschnig, M., Krautblatter, M., Lang, A., Schrott, L., and Otto, J.-C.: Current glacier recession causes significant rockfall increase: the immediate paraglacial response of deglaciating cirque walls, Earth Surface Dynamics, 8, 729–751, , 2020.

Hergarten, S.: Topography-based modeling of large rockfalls and application to hazard assessment, Geophysical Research Letters, 39, , 2012.

Hungr, O.: Prospects for prediction of landslide dam geometry using empirical and dynamic models, in: Natural and Artificial Rockslide Dams, pp. 463–477, Springer, 2011.

Hungr, O. and Evans, S. G.: Entrainment of debris in rock avalanches: An analysis of a long run-out mechanism, Bulletin of the Geological Society of America, 116, 1240–1252, , 2004.

Hungr, O., Evans, S. G., Bovis, M. J., and Hutchinson, J. N.: A review of the classification of landslides of the flow type, Environmental and Engineering Geoscience, 7, 221–238, , 2001.

Korup, O.: Geomorphometric characteristics of New Zealand landslide dams, Engineering Geology, 73, 13–35, , 2004.

Kuhn, C.: Austrian Rockslides `http://geol-info.at/index.htm`, visited 2020.07.27.

Montgomery, D. R.: Slope distributions, threshold hillslopes, and steady-state topography, American Journal of Science, 301, 432–454, , 2001.

Sanne, M. J.: Modelling the 1987 Val Pola rock avalanche using the shallow water equations, Tech. rep., Faculty of Environment and Natural Resources Albert Ludwigs University of Freiburg, 2015.

Tacconi Stefanelli, C., Segoni, S., Casagli, N., and Catani, F.: Geomorphic indexing of landslide dams evolution, Engineering Geology, , 2016.

Tebbens, S. F.: Landslide Scaling: A Review, Earth and Space Science, 7, e2019EA000 662, , 2020.

Wang, G. and Sassa, K.: Pore-pressure generation and movement of rainfall-induced landslides: Effects of grain size and fine-particle content, Engineering Geology, 69, 109–125, , 2003.

---

## Author Response (AR1)

**Authors' Response to Reviews of**

**"Controls on the formation and size of potential landslide dams and dammed lakes in the Austrian Alps"**

Anne-Laure Argentin et al.

Natural Hazards and Earth System Sciences, https://doi.org/10.5194/nhess-2020-326

RC: Reviewers' Comment, AR: Authors' Response

AR: Dear Andreas Günther,

We resubmit a revised version of our manuscript "Controls on the formation and size of potential landslide dams and dammed lakes in the Austrian Alps" to consider for publication in Natural Hazards and Earth System Sciences. First, we want to thank the two reviewers for their detailed and very constructive reviews. We appreciate their effort, which helped us to strongly improve our manuscript. We addressed almost all raised issues and revised our manuscript according to the reviewers' suggestions. Both reviewers considered our manuscript as an interesting contribution to the landslide dam community and we are confident that the revised version of this manuscript meets the high-quality standards of this journal. Before going into the details of the point by point response, we would like to emphasize the main modifications of the revised manuscript.

• As suggested by the anonymous reviewer #1, we now dedicate a whole section to each of the main limitations of the study (uniform slope thresholds, rheological model) in the Discussion. We also furthered our analysis of the impact of rheology on the landslide dam and lake geometry (Supplementary Fig. A1).

• We added a paragraph on the Austrian Alps in the Introduction as requested by the anonymous reviewer #2 and reformulated all unclear sentences and paragraphs.

• We also took into account the new work from Fan et al. (2020), which provides a new and extensive landslide dam database that we use for comparison in Figure 5. This new validation dataset fits well with our results.

As requested by both reviewers, we further performed slight modifications to the text for enhanced clarity and style.

The changes made in the manuscript can be visualized thanks to the Latex package TrackChanges. The modifications suggested by reviewer #1 and reviewer #2 are written in Blue and Green, respectively. The changes linked to the new dataset from Fan et al. (2020) are displayed in Purple while English corrections are made in Turquoise.

Thank you very much for the editorial handling.

**1. Anonymous Referee #1, Received and published: 23 February 2021**

RC: Anne-Laure Argentin et al. entitled "Controls on the formation of potential landslide dams and dammed lakes in the Austrian Alps", present a process-based modeling approach to envision susceptibility of landslide damming and lake formation by individually simulating the process chain from the initiation probability of landslides, land-slide runout, river obstruction and damming. The concept and the methods employed by the authors are thought-provoking and progressive and this manuscript would be significant for the engineering geological and natural hazard community. The idea to conceptualize the landslide dam hazard chain through a process-based modeling is appreciable. Nevertheless, the study may still need some additional elements or factors that can be considered, warranting a minor to moderate revisions to the manuscript paper to be accepted and this reviewer see the following suggestions shall be followed.

AR: We thank the reviewer for the thorough review of our manuscript and the pertinent comments formulated. We are very glad that the reviewer finds our article a significant addition to the scientific community and are confident that we addressed the issues raised in this revised version of the article. In this response we reply line by line to the suggestions made:

**1.1. Major comment #1: Modeling the landslide release areas**

- RC: The authors adopted a slope-based criterion following Hergarten et al. 2012 to determine the probability of landslide release areas. The authors do mention the reason for their choice "The approach proposed by Hergarten (2012) still seems to be the only model in this context which is able to predict the observed power-law distribution of rockfall and rockslide volumes". However, the performance of the model in a terrain with lithological variations need to be questioned. Different rocks would have different thresholds with regards to slope angle and stability.
- AR: The first method used from Hergarten (2012) is an empirical method, which is not physically process-based. Although it seems intuitive that different rocks would have different thresholds with regards to slope angle and stability, as formulated for soil-mantled slopes and bedrock slopes (Montgomery, 2001), it has seldom been tested with bedrock slopes (Goudie, 2016) and does not explain some "over-steepened" slopes (Fernández et al., 2008). Furthermore, rock mass strength is a variable that is controlled by a large set of parameters (e.g. lithology, structural discontinuities etc.), and, to our knowledge, no thresholds are available based on rock mass strength for our study area.
- **RC:** In addition, rockfalls possess strong sensitivity towards discontinuities. I would request the authors to perform a validation of their analysis of landslide probability.
- RC: Is it possible to compare the landslide probability estimated by the Hergarten et al. 2012 to actual events of landslides within different geological units of the study area? It would be nice to see the performance of the model for past cases at first and then use it to predict the future.
- AR: The reviewer is right to mention the influence of lithology and discontinuities on landslide triggering. Our method uses a statistical approach that only holds for extended regions, does not take into account lithological variations and structural discontinuities, and thus cannot be applied to reproduce case studies. Moreover, we assume that the different stability thresholds lead to an equafinality of results. Taking the same thresholds for the whole mountain range allows for a simple model. The topography is the main control of landsliding here. Furthermore, no temporal constraints are applied to our model, and we do not investigate any triggering return periods or landslide frequency. We thus call "density" and not "frequency" the number of landslide simulated per km2. This density is closer to a landsliding potential, with high densities where not much landsliding has already occurred (i.e. in steep terrains).

We validated our model by visually comparing the landslides created with a landslide database over the Austrian Alps. However, to provide a better overview of the difference between landslide densities per tectonic unit, created a histogram of the landsliding frequency in each lithological unit for both our model and the database:

The Tirolian nappes show a higher density of landslides in our model than in the datasets from Kuhn (visited 2020.07.27) (Fig. 1), which is logical since the time range investigated by the database is restrained. We notice that 1) the (computationally-driven) decision to choose landslides with  $V > 10^5$  m3 exacerbates differences with landslide database densities, 2) landslide densities are overestimated in steep terrains (e.g. calcareous nappes).

However, this model recreates the typical power-law scaling of landslides (Fig. 2, Tebbens, 2020), and changing the thresholds do not change the size distribution of landslides by much (Hergarten, 2012).